# Equids’ Core Vaccines Guidelines in North America: Considerations and Prospective

**DOI:** 10.3390/vaccines10030398

**Published:** 2022-03-04

**Authors:** Hélène Desanti-Consoli, Juliette Bouillon, Ronan J. J. Chapuis

**Affiliations:** 1DoctoraDesanti, San Jose 11801, Costa Rica; helenedesanti@hotmail.com; 2Department of Clinical Sciences, Ross University School of Veterinary Medicine, P.O. Box 334, Basseterre, Saint Kitts and Nevis; jbouillon@rossvet.edu.kn

**Keywords:** core vaccine, guidelines, equine, horse, donkey, rabies, tetanus, Eastern equine encephalomyelitis, Western equine encephalomyelitis, West Nile

## Abstract

Vaccination against infectious diseases is a cornerstone of veterinary medicine in the prevention of disease transmission, illness severity, and often death in animals. In North American equine medicine, equine vaccines protecting against tetanus, rabies, Eastern and Western equine encephalomyelitis, and West Nile are core vaccines as these have been classified as having a heightened risk of mortality, infectiousness, and endemic status. Some guidelines differ from the label of vaccines, to improve the protection of patients or to decrease the unnecessary administration to reduce potential side effects. In North America, resources for the equine practitioners are available on the American Association of Equine Practitioners (AAEP) website. Conversely, in small companion animals, peer review materials are regularly published in open access journals to guide the vaccination of dogs and cats. The aims of this review are to present how the vaccine guidelines have been established for small companion animals and horses in North America, to review the equine literature to solidify or contrast the current AAEP guidelines of core vaccines, and to suggest future research directions in the equine vaccine field considering small companion animal strategies and the current available resources in equine literature.

## 1. Introduction

In humans and animals, immunization has proven efficacious to decrease the morbidity and mortality in infectious diseases [1]. Inactivated and live-attenuated vaccines stimulate humoral responses resulting in the production of pathogen specific neutralizing antibodies. Third-generation vaccines (DNA, RNA, and recombinant viral-vector) also stimulate T-cell mediated immune responses, which are suggested to be, at least, as important as the humoral response to protect against infection [1]. Core vaccines are defined by the American Veterinary Medical Association as vaccines protecting from endemic diseases, regulated diseases for public health, highly contagious diseases, and severe diseases [2]. Core vaccines should be safe and proven efficacious [2]. While adverse events appear to be rare, every vaccine has the potential to cause a variety of responses ranging from mild and localized to severe and systemic and occasionally can be fatal [2,3]. Adverse reactions to vaccines in equine are under-reported [2,3], and poorly documented [4]. Until proven safe for any equine patient, unnecessary administration of vaccines should be avoided [3]. Guidelines are useful to the practitioners to make the decision to administer vaccines only when indicated.

In equine, vaccines protecting from tetanus, rabies, and viral encephalitis caused by Eastern Equine Encephalitis (EEE), Western Equine Encephalitis (WEE), and West Nile Virus (WNV) are considered core vaccines by the American Association of Equine Practitioners (AAEP) [5]. The AAEP provides online open access guidelines to equine practitioners regarding vaccine protocols, which can differ from the manufacturers’ label instructions [5]. These guidelines are established by the AAEP Infectious Disease Committee (communication with the AAEP). At the time of the review, the limitations of the AAEP guidelines included the lack of supporting data and references used to establish the guidelines, apart from the one article cited in the WEE chapter, as well as the lack of declaration of conflict of interest and funding sources, and dates of revision for most pages. The AAEP published reports periodically to summarize guidelines in specific targeted equine populations [3,6], however, the full guidelines were not published in the peer review literature.

In contrast, small companion animal practitioners have an open access to peer review guidelines on vaccine protocols for different populations of dogs and cats. These guidelines have been regularly updated and were built from a consensus of experts collecting evidence-based and clinical data. Practitioners have access to tables summarizing immunization protocols, recommendations regarding vaccination decisions based on serology results, and references to updated supportive data [7,8]. Suggestions have been made that vaccine guidelines in equine should be developed using similar guidelines as those used for the production of vaccines for small companion animals [3].

The aims of this review are: (1) to summarize how guidelines were established for small companion animals to serve as a base of comparison with equine vaccination guidelines; (2) to present potential limitations of the AAEP guidelines equine core vaccinations and resources that can complete these guidelines; and (3) to propose future research to improve the guidelines of equine core vaccination.

## 2. Establishment of Guidelines in Dogs and Cats

For years, small companion animal vaccination recommendations have been successively published by the American Veterinary Medical Association (AVMA), with reviews dating back to 1982 [9]. In 2000, debate was raised as to whether revaccination should occur yearly or triennially for some vaccines [10]. As a result, the AVMA Council on Biologic and Therapeutic Agents was created including experts in internal medicine, immunology, microbiology, infectious diseases, as well as medicine and clinical practice. In 2002, the AVMA’s Principles of Vaccinations and a report from the AVMA Council on Biologic and Therapeutic Agents on cat and dog vaccines were published [11,12].

Subsequently, the American Animal Hospital Association (AAHA) created a task force who established a review process based on existing literature and expert opinion. As such, in 2003, the first AAHA Canine Vaccine Guidelines were published [13]. With the advent of new vaccines and the increasing body of literature, these guidelines have been regularly updated with, at the time of the review, the most recent published by the AAHA in 2017 for dogs [8], and by the AAHA/AAFP (American Association of Feline Practitioners) in 2020 for cats [7]. Additional guidelines have been written since 2006 by the World Small Animal Veterinary Association (WSAVA) with the aim of providing global application and allowing practices to develop vaccination protocols relevant to their local conditions [14]. Therefore, in cats and dogs, two independent groups (AAHA/AAFP and WSAVA) publish peer reviewed guidelines accessible to practitioners.

Over time, all guidelines agreed on the definition of core and non-core vaccines [7]. The main changes pertained to the recommendations made for the timing of core vaccination of puppies and kittens, revaccination intervals for adult dogs and cats, information about newly available vaccines, and reclassification of vaccines to core or non-core [14]. For example, numerous serologic studies demonstrated that the canine parenteral multivalent vaccines including core vaccines were considered efficacious and safe for a triennial booster instead of a yearly booster after completion of the initial series [8]. Similarly, studies reported extended periods of immunity for feline core vaccines leading to the recommendation that boosters could be administered triennially in low-risk felines defined as solitary, indoor cats who do not visit boarding catteries [7,14].

More recently, guidelines have included recommendations differentiating categories of pets: client-owned and shelter-housed dogs as well as shelter cats, trap–neuter–return/trap–neuter–release cats, cattery cats, and foster cats. Additional information included vaccination laws and regulations (e.g., internet links for state-by-state information on rabies), vaccine storage and handling, indications for antibody testing (serology) with recommended actions based on test results, product information on the emerging class of immunotherapeutics approved for use in veterinary medicine [8], and updates regarding vaccine reactions [7]. Furthermore, the AAHA offers an open access online interactive platform that provides guidelines and updated information to practitioners and gives lay information to cat owners, on a case-by-case basis [15].

In small companion animals, serological testing is advocated to monitor vaccine responses and adapt protocols with commercial kits developed and available to the practitioners [14]. However, for practicability and economic reasons, testing for determination of protective immunity in puppies or revaccination intervals in adult animals is not done [14]. Puppy and kitten vaccination protocols have been adapted to ensure the lack of maternal antibodies interference from vaccinated bitches and queens. Vaccination of pregnant animals is not routinely recommended for most vaccines and instead recommended before pregnancy/breeding [14]. Vaccinations of newborns start at six to eight weeks old, followed by the administrations of boosters every two to four weeks until minimum age of 16 weeks. Therefore, up to five vaccines are administered before they are six months old. The most recent updates recommended that revaccination after the initial series must be performed at six months of age, instead of one year, to maximize the avoidance of the immunity gap [7,8,14]. While naive aged animals might not respond efficiently to vaccines and require specific protocols of vaccination, old animals fully vaccinated can be boostered based on serological testing [14].

## 3. Guidelines of Equine Core Vaccines

The recommendation to vaccinate broodmares hold a double objective: to protect the mare from some abortive diseases and tetanus, and to stimulate the production of IgG in the colostrum to enhance the passive immunity and protect the newborn foal [5]. Most core vaccines were not labeled for pregnant mares and evidence supporting the safety of their administration is limited. Some suggest repeating vaccination in pregnant mares might increase risk for local and systemic diseases in mares, and that mares should not be vaccinated in the first 60 days of pregnancy as fetal organogenesis occurs in early gestation [16]. However, these recommendations remain experts’ opinion, and are not supported by data with higher level of evidence. The AAEP recommended administering boosters of core vaccines four to six weeks prior to parturition [5]. However, the administration within the last two months of pregnancy of modified-live vaccine against equine viral arteritis (EVA) showed risk for abortion, while no abortion occurred when administered at mid-gestation [17]. Adverse events other than abortion (e.g., recurrent seizures) were associated with the administration of core vaccines in the broodmare [4]. Finally, many vaccines contain thimerosal, an agent that raised controversial safety concerns for the fetus both in humans and animals [3,18]. Therefore, it seems prudent to confirm the safety of administration of each vaccine to broodmares with long term follow-up, of both mares and foals, after the administration of the vaccine at a different stage of pregnancy.

When pregnant mares were vaccinated with a multivalent vaccine (West Nile Innovator + EWT) one month prior to foaling, their foals had detectable antibodies against West Nile, EEE, WEE, and tetanus after the ingestion of colostrum [19]. However, no history regarding the past exposure to natural infection was reported in these mares and no control population was included (i.e., naïve mare and mares up to date on vaccine, although not boosted prior to foaling). To the authors’ knowledge, there was no proof of the benefit of administrating a booster of core vaccines to broodmare prior to parturition; this should be investigated by comparing the concentration of antibodies in the colostrum of mares either receiving or not receiving a booster prior to foaling. When vaccinated with a modified-live virus vaccine (EVA) during late pregnancy, up to 10 weeks prior to parturition, not all mares had colostrum antibodies against EVA; conversely all mares vaccinated during mid pregnancy, 20 to 11 weeks prior to parturition, had antibodies against EVA in colostrum [17]. This suggested for some vaccines, it might be beneficial to vaccinate broodmares earlier than four to six weeks prior to parturition. The ideal time to vaccinate broodmares during pregnancy to enhance the production of colostrum antibodies in core vaccines should be investigated by comparing the concentration of antibodies in the colostrum of vaccinated mares receiving boosters at different times of pregnancy. Increased production of specific antibodies in colostrum secondary to vaccination of pregnant mares was also found with *Lawsonia* and *Rhodococcus* vaccines [20,21,22]. However, increasing the concentration of colostrum immunoglobulins by vaccinating broodmares against *Rhodococcus* did not systematically achieve a protective effect on the foal and seemed to depend on the type of vaccine [21,22]. This highlights the need to study the benefit of vaccinating broodmares to enhance colostrum antibody production and the subsequent protective efficacy in foals; this would require clinical trials exposing foals to diseases after the ingestion of colostrum from mares either receiving or not receiving a booster during pregnancy. Finally, regarding the vaccination of foals, the AAEP guidelines did not differentiate the different vaccinal status of mares updated on vaccine. Indeed, mares up to date on vaccine could either be receiving or not be receiving additional boosters during pregnancy or peripartum periods. The need to adapt vaccinal protocols in foals depending on the vaccination and booster status of the mare during pregnancy or peripartum periods should be investigated for all core vaccines.

One month old foals had similar humoral responses to two dose primary injection of a Keyhole Limpet antigen if they were born from vaccinated mares two and three months before foaling, than if they were born to naïve mares [23]. This suggested the presence of maternal antibodies did not interfere with the vaccination of foals. However, maternal antibodies, including vaccine-specific antibodies, showed suppressive effect in humoral and cell-mediated responses of neonatal foals [24,25]. Furthermore, if the same influenza vaccine was used to vaccinate mares and their foals, the foals did not mount detectable humoral immunity despite the administration of multiple injections [26]. This phenomenon, referred to as “tolerance-like phenomenon” was also reported with the immunization against EEE [2,27]. Conversely, when mares received a multivalent vaccine (West Nile Innovator + EWT) one month prior to foaling, their foals mounted a humoral and cellular immune response when vaccinated with the same product [19]. Overall, the “tolerance-like phenomenon” might not only depend on the presence of maternal antibodies, but also on vaccine/antigens administered and vaccination protocols used during pregnancy. In order to adapt vaccination protocols in foals, further studies should compare the immune responses to vaccine in foals born from mares with different vaccinal status (naïve, vaccinated with or without boosters during pregnancy), and compare the immune responses to vaccines in foals when the same antigens are used for the vaccination of the mare and the foal.

When mares were vaccinated one month prior to foaling with a multivalent vaccine (West Nile Innovator + EWT), and their foals were administered the same vaccine, a three dose primary vaccination, started at three months old, with boosters administered at four and six months old, resulted in the same humoral and cellular responses, than a two dose primary vaccination 30 days apart started at six months old [19]. A booster at 11 months old resulted in the same response in both groups [19]. This suggested that despite the presence of maternal antibodies, starting the vaccination at three months of age protected foals to a similar degree as starting at six months. However, the conclusion of Davis et al. (2015) that foals showed the same response at three months old compared to six months old is debatable [19]. Indeed, in the three month old group, despite vaccination, serum antibody concentration continued to decrease until six months old in a similar trend than in the group started at six months old. At six months old both groups had the same serum antibody titers. [19]. Therefore, in the group started at three months old, the vaccination did not increase antibody titers and, the findings might suggest no detrimental or beneficial effect to start the vaccination of foals at three months old with the presence of maternal antibodies against WNV, EEE, WEE, and tetanus. Another limitation of this study was the absence of a control group, i.e., foals without maternal antibodies against these core diseases. This merits an investigation to confirm the benefit of starting the vaccination of three month old foals not protected by maternal antibodies. If confirmed, a protocol starting as early as three months, for foals at risk and/or with unknown neutralizing antibody titers, should be considered.

In the absence of maternal antibodies, the administration of two injections three weeks apart, of a killed adjuvanted vaccine, showed that three day old foals have a significant attenuated humoral and cell-mediated immune response compared to three month old foals and adult horses, with the last showing the strongest humoral and lymphoproliferative immune responses [24]. Albeit modest, the newborns were able to mount humoral and cell immune responses. Similarly, in absence of maternal antibodies, one month old foals needed two dose primary injections of a Keyhole Limpet antigen to mount humoral response, while if initiated at three or six months old, a response appeared after the first injection, with a higher response in the six month old group. However, after the second injection, all groups mounted a similar humoral response [23]. The protocols used in the immunization of newborn foals born from naïve mares should be thoroughly investigated for all vaccines.

The AAEP were presenting the same vaccinal protocols for foals born from naïve mares or mares with unknown vaccination history. Protocols might need to be adapted based on the immune status of the mares, rather than the vaccinal history of the mares. Passive immunity in foals is also variable as it depends on the quantity of colostrum ingested, window time of ingestion, and other mare parameters such as age, onset of premature lactation, or poor production [23]. Therefore, to improve the medical care of neonates, there is an obvious need for the development of diagnostic tools that could help in the decisions of early vaccination depending on the risk of exposure to the disease. Further studies should aim to develop stall side tests measuring the concentration of specific antibodies in the colostrum or the blood of either mares or neonate foals.

As reported in other species, elderly horses experience immunosenescence [28,29]. The AAEP recommendations were not presenting adapted protocols for this population. The difference of humoral response to influenza and rabies (and possibly EHV-1) vaccines were studied in old horses, and showed a lower response compared to young horses [28,29]. To the authors’ knowledge, immune responses to other vaccines in adult horses with different age categories had not been investigated. Furthermore, old horses are at high risk for Pituitary Pars Intermedia Dysfunction (PPID), which is associated with immunological deficiency [30]. The concentration of α-MSH was not found to be a risk factor associated with age difference in humoral response to vaccines [28], however the response to vaccines in horses diagnosed with PPID had not been investigated and merits further study. Selenium and vitamin E were not found to be risk factors for lower immune response in old horses compared to young horses, however, were not entirely disregarded [28]. As selenium and vitamin E supplementation could boost the humoral response [28], further studies are warranted to confirm these findings, and to investigate whether or not the supplementation of vitamin E and selenium could be a beneficial immunologic booster in the vaccination of populations with diminished immune response, such as foals and elderly horses.

### 3.1. Tetanus

Table 1 presents the AAEP guidelines, labelled protocols, and some consideration for protocols based on the available equine literature of the tetanus vaccines available in 2021 for adult horses, broodmares, and foals. Figure 1 summarizes all these data for considerations of equine practitioners while vaccinating equine patients against tetanus. Table 2 and Table 6 present the list of some labelled vaccines protecting for tetanus in North America.

Titers of tetanus toxin binding antibodies above 0.01 IU/mL are considered protective [33]. However, this level is extrapolated from human literature, itself derived from guinea pig research [34]. Horses might be protected at a lower level; indeed, titer levels as low as 0.0025 IU/mL have been shown to be protective in horses from subcutaneous inoculation of three times the lethal dose of tetanus toxin [34,35]. The protective level of neutralizing antibodies still needed to be confirmed in horses. Furthermore, a challenge clinical study suggested that a protective cellular immunity might participate in the protection against tetanus [33,35]. The presence and extent of protection of cell mediated immunity following vaccination against tetanus should be investigated, as third generation vaccines could be developed to enhance the protection from tetanus.

With a detection limit set at 0.1 IU/mL, ten times above the suspected level of protection, a stall side test is available to veterinarians in Europe (Fassisi^®^ TetaCheck Gesellschaft für Veterinärdiagnostik und Umweltanalysen mbH, Germany). Results of the test appeared to be reliable [36] and useful for research or adapting vaccine protocols in adult horses and foals [31]. While not available in North America, further research should be conducted regarding the development of such tests and their use to develop strategies of vaccination or ensure proper protection following vaccinations or birth. Nevertheless, measuring titers is available to equine practitioners in North America and can be used [33].

Some horses thought to have been vaccinated have contracted tetanus and died [37]. Another retrospective study reported that adult horses have contracted tetanus despite appropriate schedules of vaccination reported by owners [38]. Explanations other than erroneous reports from owners included high infective pressure, lower vaccine response of the horses, and/or inappropriate storage or administration of vaccines [38]. This suggested a need to pursue research on tetanus vaccination to ensure full protection of vaccinated horses, however, this would require a clinical trial and such studies can be limited due to welfare concerns.

A study showed that following two dose primary vaccination, four weeks apart, some horses might not be fully protected 16 months later. The limitation of this study was the limit of detection of antibodies at 0.04 IU/mL, and the lack of sample analysis between two weeks and sixteen months after primary vaccination [34]. Therefore, the exact period when titers remained above 0.01 IU/mL after the primary vaccination was unknown. The interindividual variation in humoral response to tetanus toxoid should be evaluated to adapt protocols in animals that lack a immune response to the vaccine.

There is evidence tetanus humoral response was long lasting in horses, which led to the suggestion to revise the guidelines regarding tetanus prophylaxis. Some studies supported that the period between tetanus boosters could be extended, however, some are not reported in English or are not accessible to the authors [33]. Some countries extended the period, including Sweden (biennial booster), the UK (triennial booster) or New Zealand (quinquennial booster) [34]. Some references suggested boosters every eight to ten years in horses [31,33]. For example, a challenge clinical study supported three intramuscular doses of tetanus toxoid (six to twelve weeks apart and a year later) being protective for eight years, and possibly for life, even if antibodies were not detected, suggesting a protective cellular immunity [33,35]. A study showed that five to eleven month old horses receiving a two dose primary vaccination four weeks apart and a booster 15 to 17 months later (Equilis Prequenza Te), maintained titers above 0.04 IU/mL at least three years [34]. However, at least one horse had titers lower than 0.04 IU/mL, which suggested the protocol would need additional investigations or revisions before recommendation. The single administration of an oil based tetanus vaccine to 10 adult horses induced titers above 0.01 IU/mL for at least 116 weeks (2.1 years) [39]. When the booster was administered at 128 weeks, a strong humoral response ensued and was detected three days post injection (>32 IU/mL at 18 days) and the two horses sampled 3.5 years later had titers above 4.99 IU/mL, supporting that long lasting humoral response against tetanus was achievable [39]. These findings supported extending the period of vaccination booster being an option in horses and should be further investigated by serological studies. With a ISCOMatrix adjuvant vaccine (Equilis Prequenza Te, 40 Lf/dose) administered to foals, two doses administered four weeks apart induced titers above 0.1 IU/mL a week after the first injection, above 20 IU/mL immediately after the second injection and remained above 0.3 IU/mL at 61 weeks (14 months) after the last injection. If a third dose is injected 22 weeks (five months) after the second, the titers immediately reach above 110 IU/mL and remain much higher than 0.02 IU/mL at 103 weeks (23 months). Therefore, protective titers were maintained for at least 18 months after a two dose primary injection and at least 24 months if a third injection is provided at five months [40]. Further studies should be carried out to confirm the benefit of a three dose primary vaccination followed by an extended time between boosters.

Strong and immediate responses to boosters were reported in previously immunized horses [39]. This supported the benefit of the AAEP recommendation to vaccinate animals at risk of developing tetanus from wounds or surgery to reach high titers at the onset of the disease. Indeed, reported cases of tetanus followed histories of wounds within an average of nine days prior to the onset of disease (range 2–27 days) [37]. Conversely, titers above 0.01 IU/mL after the first injection of a Purified Aluminum Phosphate vaccine (10 Lf/mL—2 mL, two doses four weeks apart) were obtained at 1.5 weeks in eight to twelve month old horses [41], which can be too late when an immediate protection is required. This leads to the recommendations for not only a primary vaccination, but also to administer tetanus antitoxin to patients at risk of tetanus. Combined active–passive immunity was proven beneficial to ensure circulating antibodies above 0.01 IU/mL when administering a Purified Aluminum Phosphate vaccine (10 Lft/mL—2 mL, 2 doses four weeks apart), and antitoxin at opposite side of neck (1500 units) in eight to twelve month old horses [41]. A slightly lower concentration of antibodies was reached in the group administered vaccine and antitoxin compared to antitoxin only, while titers were higher in the combined group compared to the only vaccinated group [41]. Horses weighed 340 kg in this study. It seems wise to pursue further studies to confirm that a protective level of serum antibodies in heavier horses would be achieved and to confirm the absence of interaction between vaccine and antitoxin in lighter horses.

The vaccination of pregnant mares against tetanus (Encevac TC-4 with Havlogen, or Fluvac EWT) two months prior to foaling might increase the concentration in immunoglobulins in colostrum, however due to the lack of a control population (unvaccinated mares), this was not proven [25]. Regardless, if no direct correlation was found between the serum titer of the mares previously vaccinated against tetanus (unknown vaccine) and their foals after colostrum intake, mares with high titers had a tendency to have foals with high titers after colostrum intake and vice versa [39]. Further studies should be provided to confirm the benefit of tetanus vaccination to enhance passive immunity, and to investigate the best window of vaccination of broodmares to enhance the production of antibodies in colostrum.

Horses younger than one year old mounted a lower immune response following multivalent tetanus vaccines (Encevac TC-4 with Havlogen, or Fluvac EWT) than yearlings [25]. If maternal antibodies were present, some six to seven month old foals had a lower humoral response to vaccine (Equilis Prequenza Te) compared to six to eight month old foals without detected maternal antibodies. It is unknown if the mares of these foals were vaccinated during pregnancy [34]. When maternal antibodies were detected in foals, no foals mounted humoral response when vaccinated at 10 to 18 weeks old (three to five months old) with an oil based tetanus vaccine; they did mount a response with a second injection twelve weeks (three months) later, but some may have required a booster as titers below 0.01 IU/mL were detected thirteen weeks (three months) after the second injection [39]. This supports the need for multiple injections in foals, especially with the presence of maternal antibodies, however the protocol might differ depending on the type of vaccine and the maternal antibody concentration. Vaccination of pregnant mares against tetanus (Encevac TC-4 with Havlogen, or Fluvac EWT) two months prior to foaling caused failure from the foals to mount immunity if vaccination was started at three months old. The responses to vaccines were evident when vaccinations were started at six months old, which lead to the recommendation to not vaccinate foals from immunized mares before six months [25]. This contrasts with the AAEP recommendations.

Tetanus has been reported in equines as young as six days old. Success of passive immunity was not reported in these foals, and the vaccination status of the dam was unknown [38]. Nevertheless, being able to assess the appropriate humoral protection in the newborn foal might prompt further studies. In addition, the administration of tetanus antitoxin after colostrum intake in newborns confirmed with a lack of tetanus protection might be valuable, and future research is desirable to guide the practitioner to make decisions.

Finally, recovering from tetanus is considered as efficient as a primary vaccination by AAEP guidelines. This approach might be unreliable as there is no definitive ante-mortem diagnosis for tetanus, and a recurrent case of tetanus has been highly suspected in an equid patient [32]. It might be prudent to assess the humoral response in recovered patients before assuming that protection is well achieved.

### 3.2. Rabies

Table 3 presents the AAEP guidelines, labelled protocols, and some consideration for protocols based on the available equine literature of the rabies vaccines available in 2021 for adult horses, broodmares, and foals. Figure 2 summarizes all these data for considerations of equine practitioners while vaccinating equine patients against rabies. Table 4 and Table 6 present the list of some labelled vaccines protecting against rabies in North America.

A murine study, highlighted the development of neutralizing antibodies, was key in the protection against rabies [44]. A protective titer has not been established for horses, therefore titers of 0.5 IU/mL are considered protective by extrapolation from studies in humans [33]. The protective effect of titers still requires validation in horses. In murine studies, cell-mediated response to rabies infection was found in vitro [45], however, this might be inhibited by rabies infection in vivo [46]. Even if a DNA vaccine against rabies generated a comparable serological response to conventional vaccines [47], the protective effect of cell-mediated response following vaccination with third generation vaccines against rabies in horses should be investigated.

Rabies was reported in vaccinated horses, most of which were younger than two years old [6,48]. Interference of maternal antibodies was demonstrated in puppies, although not in horses [49]. In horses, interindividual variability response to the rabies vaccine and interaction of maternal antibodies should be investigated.

Following the vaccination (Imrab 3) of naive horses or horses with low titers prior to vaccination, some were defined as poor responders by having titers below 0.5 IU/mL by one year post vaccination, with some horses that did not mount titers above 0.5 IU/mL after vaccination [42]. This suggested an interindividual response to the rabies vaccine and the need for adapting protocols for these horses; assessing post vaccination titers might require attention.

Previously vaccinated horses with titers above 0.5 IU/mL prior to administration of the vaccine, maintained titers above 0.5 IU/mL for two to three years [42]. This suggested annual revaccination might not be warranted and requires further studies, although this would require a clinical trial and such studies can be limited due to welfare concerns.

The interaction of maternal antibodies and foal vaccination was suspected, with most of the three month old foals not mounting a humoral response to vaccines when maternal antibodies were present. Six month old foals with maternal antibodies also seemed not respond to a first dose, however, did respond to a second dose four weeks later (7 months old) [6]. Recommendations from these findings included initiation of vaccination at three months of age in foals not born from immunized mares (two injections recommended), and six months of age when foals are born from immunized mares. This data had to be published [6], and contrasted with other opinions suggesting to not start vaccination against rabies before nine months old in foals born from mares immunized prior to foaling [16]. Overall, these contrasted with the AAEP guidelines that recommended starting vaccination as early as four months old for both groups. Further research regarding vaccination of foals and protocols for pregnant mares are warranted.

No difference was found between the immune response to vaccination in horses younger or older than 20 years old. However, the age of poor responders or the diagnosis of PPID was not reported. In addition, the potential effect of dividing horses by different ages was not reported [42]. Indeed, a tendency for lower response from horses older than 20 years old was shown compared to a group of four to twelve year olds, however, no association was found between rabies titers and α-MSH serum concentration, which suggests that PPID might not be involved in such a difference [29]. Furthermore, only 50% of young naïve horses and 18% of old naïve horses had titers above 0.5 IU/mL by 24 weeks after the administration of a unique dose of rabies vaccine [28,29]. Further serological studies could aim to confirm the absence of difference in the response of horses with age, as well as to investigate the risk factors for a poor response to rabies vaccines. Rabies, being a fatal disease with potential risk of zoonosis, a unique dose for primary vaccination of adult horses as recommended by the AAEP is likely debatable [42]. The administration of four weeks apart two dose primary vaccination of rabies (Imrab) elicited a significant booster humoral response in old (>20-year-old) and young naïve horses [29]. Such protocols could be implemented in high-risk populations and further research should investigate the benefit of two-dose primary vaccination. Furthermore, in this study, almost one third of old horses did not mount humoral titers above 0.5 IU/mL by 24 weeks after the administration of the two dose primary vaccination [29]. Further studies should investigate if better efficacy could be obtained by modifying the period between the two injections of primary vaccination or if a three dose primary vaccination would be beneficial in naïve old horses. If confirmed, this highlights the need for revision of the AAEP recommendations regarding primary vaccination for rabies in adult horses.

Further research is warranted regarding diagnostic tools to ensure proper protection and adaptation of protocols for rabies vaccination in horses. Equine practitioners in North America have access to laboratories to evaluate the humoral response in horses after vaccination [33].

### 3.3. Equine Arboviroses

Alphaviral equine encephalomyelitis include EEE virus, WEE virus, Venezuelan equine encephalomyelitis (VEE) virus, Ross River virus, and Getah virus [50]. Although EEE, WEE, and VEE viruses are related, the three viruses are genetically and antigenically distinct. Only EEE and WEE vaccines are core vaccines in the US [5]. Flaviviral equine encephalomyelitis include Japanese Encephalitis Virus, WNV, and Murray Valley encephalitis virus [50]. Eight lineages of WNV are identified, with at least lineage 1 and 2 viruses being pathogenic in horses [51,52]. Only lineage 1 virus is included in the WNV vaccines [51,52,53].

These equine arboviruses are transmitted by different *Culex* spp. and other mosquitoes, which led the recommendation to administer a booster to horses against these diseases prior to the active vector season [5,6,50]. However, no specification regarding the minimum time to observe between updating horses with vaccination against these diseases and the beginning of the active vector season were available in literature. Studies investigating the time frame between the administration of vaccines and the status of protection should be pursued, however this would require a clinical trial and such studies can be limited due to welfare concerns.

#### 3.3.1. Eastern (EEE) and Western Encephalitis (WEE), Venezuelan Encephalitis (VEE)

Table 5 presents the AAEP guidelines, labelled protocols, and some consideration for protocols based on the available equine literature of the EEE/WEE vaccines available in 2021 for adult horses, broodmares, and foals. Figure 3 summarizes all these data for considerations of equine practitioners while vaccinating equine patients against EEE/WEE. Table 6 presents the list of some labelled vaccines protecting against EEE/WEE, Tetanus, Rabies, and West Nile in North America.

Immune interference between these encephalitis viruses was suspected as horses vaccinated with a VEE vaccine had decreased humoral response if antibodies to EEE or WEE were present [56]. Titers might correlate with protection against EEE and WEE [57], however reports supporting this statement were scarce. It was suggested that titers above 1:100 are protective for EEE and WEE; if titers were lower than 1:10, the patient should be re-vaccinated; and if titers were between 1:10 and 1:100, the patient was considered with intermediate susceptibility to EEE and WEE and no consensual guidelines regarding vaccination were established [33]. Following an experimental inoculation of EEE and WEE viruses, the humoral protections mounted against WEE or EEE, after a subsequent inoculation of VEE virus, were 60% and 100%, respectively [58]. The average titers against EEE was 1:575 (range: 1:128–1:8192) [58]. If humoral response correlates with protection, this suggested that titers as low as 1:128 against EEE are protective against VEE. Cross-protective immunities from EEE and WEE to VEE were also suspected in field observations, however, contradictory results were reported [58]. In donkeys, it seems that immunization against EEE and VEE can protect from WEE [58]. Further studies should be completed regarding protective levels of titers against EEE and WEE and to assess the cross protection between these viruses, although this would require a clinical trial and such studies can be limited due to welfare concerns. If confirmed, as EEE and WEE are always combined in the vaccines, protective titers might be cumulative.

Some horses have developed alphaviruses encephalomyelitis despite vaccination [33]. Horses have died from EEE infection despite being vaccinated within the year of the onset of the neurological disease [54]. This emphasized the need to pursue research in the efficacy of vaccination against EEE and WEE.

Current inactivated vaccines might be efficacious after a two dose primary vaccinations and yearly booster, however data supporting this statement was scarce and difficult to access [33,59]. Following two-dose primary vaccination three weeks apart of a tri-valent vaccine, horses were protected from challenge at 12 months, prior to the booster [59]. However, following a booster with a killed virus against EEE, WEE, and tetanus (Equiloid), horses that were vaccinated every six months against EEE and WEE for several years showed wide interindividual variability in the humoral responses. EEE and WEE titers peaked at two weeks after the booster, decreased, and stabilized until 20 weeks, and finally decreased again at 24 weeks (six months) to reach the level of pre-booster titer [54]. This led to the recommendation to vaccinate horses at least every six months [54] and correlated with the previous recommendation of vaccination in adult horses in Florida [55]. Further studies should be pursued regarding protocols of vaccination against EEE and WEE. Some horses did not respond to vaccination [54]. Further studies should be pursued regarding the need of additional boosters in such patients. Finally, if titers should not be used to decide on vaccination administration [54], it would be wise to confirm humoral response after the administration of vaccine. Additionally, the usefulness of serology to decide on vaccination should be further investigated.

When mares were vaccinated against EEE (unknown product) three weeks prior to foaling, titers were similar in the colostrum compared to the serum (1:40 to 1:160) [55]. No control mares were included, so confirmation of the enhanced effect of passive immunity by booster was required. Interestingly, some mares had titer of less than 1:100 despite the injection during pregnancy.

Foals born from mares that were vaccinated against EEE (EEE/WEE: Encephaloid) three weeks prior to foaling had serum titers reflecting colostrum titers [55]. Furthermore, after adequate colostrum intake, foals had the same serum titers as their dams [60]. These suggested that administering a booster in the mare prior to foaling was able to enhance passive immunity against EEE and WEE.

The maternal antibodies interfered with the foals ability to mount humoral responses against other alphaviruses [60]. This remained to be confirmed in EEE and WEE vaccination.

If titers of maternal antibodies remained above 1:20, foals vaccinated (Encephaloid) at three months old did not mount humoral response for EEE [55]. This suggests that interference from maternal antibodies exists. However, all foals with titers below 1:10 did show an immune response. Following the second injection at four months old, not all foals had detectable humoral response, and no foal reached titers of 1:100. Finally, following a third injection at 10 months old, the average of titers reached above 1:100, however, there was wide interindividual variation between foals [55]. Humoral response to WEE had a similar trend [55]. This highlighted the necessity to pursue additional research on this topic. At that time, the necessity of assessing passive immunity to initiate vaccination at three months was suggested [55], however, no recommendations were provided as to how to adapt the protocols based on all possible results of analysis of passive immunity and decision on adapting the vaccinal protocol. Nevertheless, equine practitioners in North America can submit EEE and WEE titers analysis [33]. There is an obvious need for additional research regarding immunization of foals against EEE and WEE.

#### 3.3.2. West Nile Virus (WNV)

Table 7 presents the AAEP guidelines, labelled protocols, and some consideration for protocols based on the available equine literature of the WNV vaccines available in 2021 for adult horses, broodmares, and foals. Figure 4 summarizes all these data for considerations of equine practitioners while vaccinating equine patients against WNV. Table 6 and Table 8 present the list of some labelled vaccines protecting against WNV in North America.

Titers of 1:5 have been suggested protective, however, this is derived from hamster data [33], and the use of an ELISA kit to monitor the IgG titers against WNV was recommended to ensure adequate protection in horses [61]. A month after a two dose primary vaccination of an inactivated vaccine, titer responses ranged from 1:320 to 1:40 and none of the vaccinated horses developed clinical signs, while evidence of exposure was serologically demonstrated. First, this confirmed interindividual variability in response to the vaccine; second, this suggested that titers as low as 1:320 are protective [51]. Similar findings were reported following two dose primary vaccination 35 days apart in adult horses of recombinant WNV vaccine (RECOMBITEK^®^): two weeks after the second injection, titers ranged from 1:118 to 1:31, and no horses developed significant clinical signs of WNV after intrathecal virus inoculation, suggesting that titers as low as 1:118 were protective. Viremia was also controlled with this vaccine [62]. Following a single injection of a chimera vaccine (Prevenile, Intervet), titers ranged for 1:128 to 1:4 a month after injection, at 12 month titers ranged from 1:32 to 1:8 in all except one horse, which was the only one to develop signs consistent with West Nile following intrathecal challenge [63]. This suggested that protective titers could be as low as 1:32. Two weeks after a two dose primary vaccination four weeks apart of another third generation vaccine (ALVAX-WNV), titers of neutralizing antibodies ranged from 1:160 to 1:5 just prior to an intrathecal inoculation of a WNV. Most vaccinated horses mounted an anamnestic response quicker than the control group, no histopathological abnormalities were detected, only one horse had a low viremia while others had no viremia, and no horses developed clinical signs consistent with West Nile disease [53]. This supported the suggestion that titers of 1:160 could be protective. However, third generation vaccines also induce cell mediated response, which is likely involved in the protection of WNV [1,64,65]. Indeed, in an experimental study, following a single injection of recombinant vaccine (RECOMBITEK^®^ WNV), horses that did not have detectable neutralizing antibodies were also protected against a mosquito challenge [64]. Another consideration was that experimental challenges might not represent the natural infection. High numbers of infected mosquitoes feeding on a horse can likely inoculate a higher infective dose in some natural context [3]. In an experimental challenge, only one of nineteen horses was detected with transient viremia, and no horses, vaccinated or control, developed clinical signs of West Nile disease [66]. Further studies were necessary to confirm the clinical efficacy of West Nile vaccines and the protective level of neutralizing antibodies, although this would require a clinical trial and such studies can be limited due to welfare concerns. A third consideration was titers were determined by seroneutralization tests, which do not differentiate between the different neutralizing antibodies, IgG and IgM [67]. IgM detection is indicative of recent exposure to the virus [67]. Vaccinated horses were believed to not develop IgM response to vaccine, demonstrated with the administration of whole killed vaccine (RECOMBITEK and Vetera) [61], however, some references have suggested otherwise [68,69]. The authors did not find evidence that supports the last statement and further investigation should be pursued. Finally, the duration of IgM response was unknown in horses, although it was suspected that IgM were detectable for less than three months [70]. The short duration of IgM was also assumed in humans, and it was found that IgM could persist for about 500 days in some patients [71]. Therefore, evaluating serum titers of neutralizing antibodies in vaccinated horses exposed to the virus might overestimate titer levels, and comparing protective titer levels between vaccine studies holds some limitation. An investigation to confirm the protective level of neutralizing antibodies is required, however, this would require a clinical trial and such studies can be limited due to welfare concerns.

Labeled vaccines have been developed with lineage 1, although cross protection was demonstrated against lineage 2 strains. Indeed, field studies showed following the administration of an inactivated lineage 1 vaccine (Equip WNV), synthesis of neutralizing antibodies against lineage 2 occurred at a slightly lower concentration than lineage 1 [51,52]. In addition, none of the vaccinated horses developed clinical signs, while evidence of an exposure was serologically demonstrated [51]. Similarly, an experimental study of a third generation lineage 1 vaccine (modified live recombinant canarypox virus—ALVAC^®^-WNV, same than RECOMBITEK^®^ WNV), showed effect from intrathecal inoculation of lineage 2 virus by significantly reducing viremia post inoculation and appearance of severe clinical signs [53]. Studies should be pursued in other WNV vaccines to confirm these findings for other lineage 1 vaccines. Interestingly, horses vaccinated against other Flavivirus could mount an immune response against WNV, which suggested the existence of cross protection between Flavivirus [72]. This highlighted the fact that lineages of WNV are differentiated based on phylogenetic (i.e., molecular difference) [51], which is different than antigenic differentiation. Further investigation to confirm cross protection and the protective level of the neutralizing antibodies that provides the cross protection would be valuable, especially for countries in Asia where other Flavivirus vaccines are available and where the risk of introduction of WNV exist [72].

The administration of a killed virus vaccine, at different dosage of antigens, 3 to 4 weeks apart, induced humoral response that was still detectable 1 year later in the tested group of horses (geometric mean titer of 1:14, 2/17 horses with no detectable neutralizing antibodies). Following experimental challenge of these horses, only one of 19 horses was detected with transient viremia, and none of the horses developed clinical signs [66]. This suggested a long-lasting protection after a two dose primary vaccination. Following a two dose primary vaccination (3 or 4 weeks apart) against WNV with an inactivated lineage 1 vaccine (Equip WNV), the IgG response was mounted and declined within a few months but remained detectable until the yearly booster. Compared to the primary vaccination the yearly booster induced a stronger response, which might be long lasting. Subsequent yearly boosters might not be necessary as titers remained above 1:100 [52]. Therefore, it seemed valuable to add a third injection for primary vaccination, 4 months after the two dose primary vaccination to induce a long lasting protection [52]. This protocol should be investigated to explore options for extending the revaccination intervals against West Nile as it has been suggested as a potential option [33].

A retrospective study supported the safe use of a killed WNV vaccine (Innovator) in broodmares, but this study focused only on pregnancy outcome [73]. The study held several limitations and did not rule out the potential link of West Nile vaccination with abortion as well as deformity in foals as previously reported [2,73]. There was evidence of transfer of antibodies against West Nile virus in colostrum from dams naturally exposed to the virus. Ingestion of adequate quantity of colostrum resulted in the same titer in foals compared to their mares. Foals might be adequately protected against WNV [74]. When WNV booster of killed virus (West-Nile Innovator) was administered to mare 30 days prior to expected foaling, foals had higher titers against WNV compared to the control group. The control group and the treatment group both received the West Nile vaccine prior to breeding. Regardless if IgG against WNV in colostrum was not measured, this highly suggested an effect in boosting IgG production against WNV [19]. The foals vaccinated around 7 months of age, mounted a similar humoral response to the vaccine than the control foals, which suggested the absence of maternal antibodies interference at this age [19]. Interestingly, mares were also administered tetanus, EEE, WEE, VEE, and EHV vaccines prior to foaling [19], and no report presenting immunological response towards these diseases was found by the authors. Overall, investigations regarding the effect of maternal antibodies in foals vaccinated at 4 to 6 months old, as recommended by the AAEP, were still lacking.

Following the administration of the 1st dose of a killed virus vaccine (Innovator, Fort Dodge Animal Health), 44% of horses did not have detectable serum neutralizing antibodies; 14% did not 4 to 6 weeks after 2nd administration, and horses older than 10 years (mean of 16 years old) had lower response [65]. This highlighted the wide interindividual response to West Nile vaccine and the need for specific protocol to the elderly patients.

There was evidence that horses recovering from the West Nile disease had higher titer response after onset of the disease, lasting longer compared to vaccinated horses after the administration of vaccine [65]. In addition, unvaccinated horses exposed to natural infection did not develop the disease systematically, however, could mount an immune response [75]. Investigation to adapt the vaccine protocol applicable to exposed animals or recovered animals warrants further research. Finally, some vaccinated horses did not developed an anamnestic IgG response when exposed to WNV [53]. The equine immune response to WNV might require further study to help develop vaccines, although this would require a clinical trial and such studies can be limited due to welfare concerns.

## 4. Other Considerations

Almost 20 years after this was acknowledged, it appeared in 2021 there was still a need for prospective trials of vaccine efficacy and safety in equines [3]. However, experimental or field studies on core vaccines could be unethical to pursue. Nevertheless, there was a lack of data to provide precise recommendations on vaccine protocols. Investigating the vaccines is also challenging considering the different definitions of protection and efficacy. For example, vaccine objectives can be to protect for the development of severe clinical signs, while others can be to prevent solely for viremia, which can occur with a low infective dose and does not lead automatically to the development of clinical signs [3]. Another area that needs further research is the interference from vaccinal antibodies while testing vaccinated horses to specific diseases [3]. The development of the third generation Differentiated Infected from Vaccinated Animal, or DIVA, vaccines can be a solution [1], and the development of diagnostic tools differentiating between vaccinated and naturally infected animals [33], might facilitate field research on efficacy and safety of vaccines.

While vaccinal protocol could be adapted based on titer levels, as is the case with small companion animals, investigation into whether protocols could be adapted based on titers after recovery from natural infections would also benefit the vaccine guidelines. Indeed, immune response to natural infection is different than from vaccination [3], and the measurement of titers is recommended to adapt protocols before vaccinating against Strangles in horses [76]. This highlights that adapting protocols based on titers can be achieved for some diseases in equine. Alternatively, studies showed vaccination against tetanus elicited a strong acute phase response [77], and vaccination against WNV changed the serum proteins and leukocyte count profiles [78]. Further studies should be pursued to investigate the benefit of using the acute phase proteins and cell count as biomarkers to assess the immune response from vaccination.

The small companion animal guidelines provide specific recommendations depending on the environment where the cat or dog live, including shelter, client-owned, or foster, and an online platform has been developed to provide individual vaccine guidelines to practitioners in cats. These might be worth investigating and developing for horses.

Interestingly, both in small companion animals and equine, there are no reports about the effect of plasma transfusion and the subsequent interaction with vaccination. This might be worth further investigation. Indeed, the presence of antibodies has been shown to interfere with vaccination, plasma contains antibodies and plasma transfusion is a common practice in neonatal equine medicine.

The overall effect on equine humoral response after the use of multivalent vaccines or the combination of several vaccines should also be further investigated. Indeed, while there is interindividual variation in the response to vaccines, there are reports that show a variability in antibody response against specific antigens between different multivalent products [79,80]. These variations were suspected from antigen load and adjuvant formulation [80]. In addition, when a same valence (WNV) was injected separately as monovalent or as a combination in multivalent vaccine, the humoral response towards WNV was higher than when administered separately as monovalent, despite the same antigen mass in a multivalent product [81].

While donkey medicine is in need of pharmacological research [82], vaccine research seems imperative as well [83]. Donkeys and mules might have different susceptibility to infectious diseases compared to horses [58]. Therefore, potential different immune responses to vaccines might exist in donkeys compared to horses, and the titers to reach for protection against diseases might also be different to horses. This might be true for all other equids species, and the AAEP calls for the decision of the veterinarians for vaccination in these species [5].

Finally, the list of core vaccines differed per country. In Europe, the equine practitioners had scarce information regarding core vaccinations. The Horserace Betting Levy Board is a statutory body and serves as important reference for equine practitioners. Practitioners could access these recommendations online or at the free Levy Board application: EquiBioSafe. These resources include suggestions common to France, Germany, Ireland, Italy, the United Kingdom, and other European countries focusing on the horseracing and breeding populations, along with exportation of horses. The application, similar to that available to practitioners to vaccinate cats, gave some personalized advice based on the vaccinal status of the horse, however, seemed outdated in 2021. Nevertheless, such tools would be valuable to develop for the core vaccination in horses in the USA. Suggestions based on non-contagious infectious diseases such as tetanus or rabies are not included in their guidelines given the low risk of affecting their equestrian events. Since rabies is not entirely controlled in Europe [84], practitioners should be prudent and follow up on any new cases in their areas so they can recommend horse vaccination against the disease. The Levy Board included suggestions for the breeding population, such as the vaccination of stallions and teasers for EVA, and all breeding animals for EHV to prevent abortion. To prevent respiratory diseases, horses should be vaccinated for Equine Influenza Virus and EHV. Strangles should be considered on a base-to-base case in conjunction with the veterinarian since the immunization would interfere with diagnostic tools. In addition, WNV is considered a risk-based vaccine, hence horses should only be vaccinated if travelling to an endemic area.

Regarding other equestrian activities, the Fédération Equestre Internationale is the world governing body for Jumping, Dressage and Para Dressage, Eventing, Driving and Para Driving, Endurance and Vaulting and this organization also provide regulation regarding vaccination status of the horse.

European practitioners have limited access to vaccination guidelines. To mention a few examples, by the end of 2021, vaccine guidelines were not available to equine practitioners on the Association Vétérinaires Equine Française, or on the French Veterinary Medical Association. Similarly, in the UK, the National Office of Animal Health recommended the vaccination against tetanus and equine influenza. Finally, in Germany, the guidelines Leitlinie zur Impfung von Pferden serves as the reference, and include tetanus, equine influenza, equine herpesvirus type 1 and 4 as core vaccines [31].

## 5. Conclusions

There is a need for research to improve the vaccination guidelines and protocols in equine. The main topics of research highlighted in the current review include: (1) vaccine safety; (2) the effects of maternal antibody interference and the vaccination of foals; (3) the development of vaccination guidelines in elderly horses and the investigation of the effect of PPID in immune response to vaccine; and (4) the confirmation of the protective titers for each vaccine, and subsequently the development of serological testing to adapt vaccine protocols.

In conclusion, while guidelines continue to be developed, the authors recommend following the AAEP guidelines and the product label, with special attention to be taken when multiple vaccines are administered, or for meat withdrawal time when applicable, as well following all the mandatory vaccines the statutory organization guidelines have deemed necessary in order to participate in specific equestrian activities. Finally, it is important to communicate to owners that no vaccines have 100% efficacy in protecting against clinical disease, and nor are they 100% safe. However, the risk/benefit balance is greatly in favor of vaccination. Due to the existence of limitations regarding the guidelines available to equine practitioners, the full consent of the owner might have legal merit when vaccines are used off label.

## Figures and Tables

**Figure 1 vaccines-10-00398-f001:**
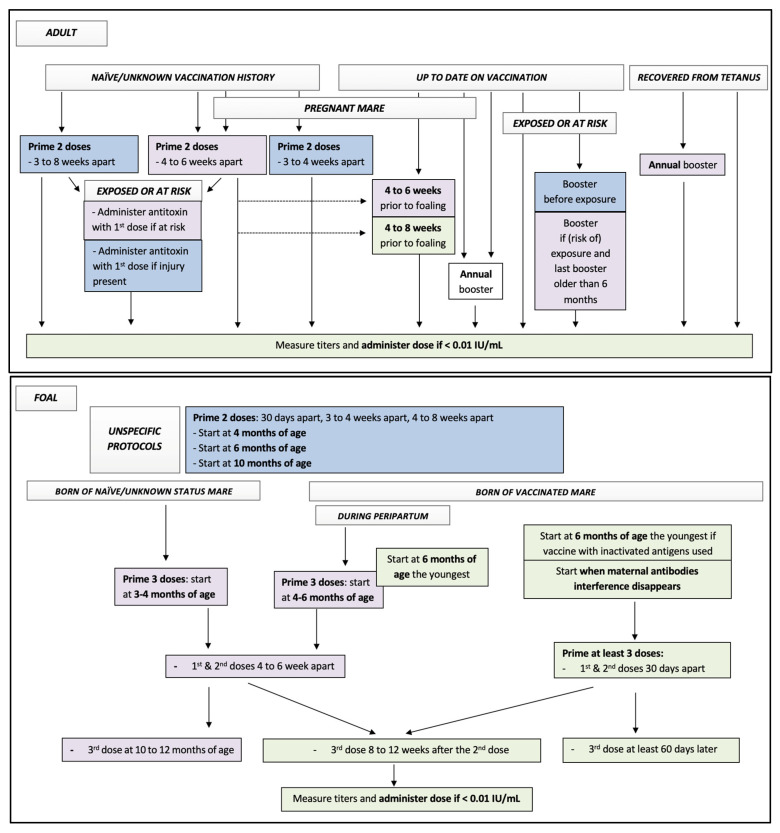
Flow chart presenting all available data to equine practitioners regarding decisions on the protocol of vaccination against tetanus in horses. Key: blue squares, data from all labelled protocols; purple squares, data from AAEP recommendations in 2021; green squares, data from equine literature. See tables for details regarding supportive data and specific labeled protocols.

**Figure 2 vaccines-10-00398-f002:**
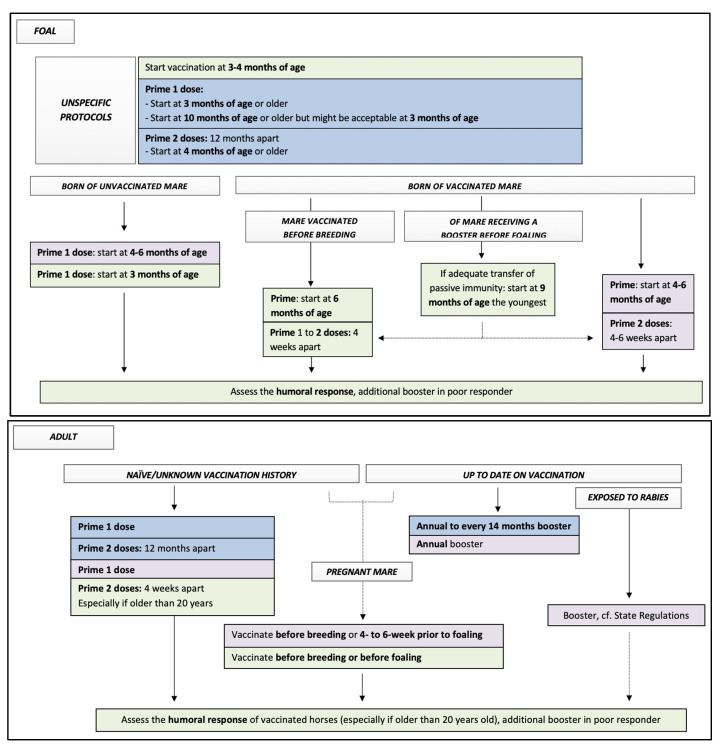
Flow chart presenting all available data to equine practitioners for decision on protocol of vaccination against rabies in horses. Key: blue squares, data from all labelled protocols; purple squares, data from AAEP recommendations in 2021; green squares, data from equine literature. See tables for details regarding supportive data and specific labeled protocols.

**Figure 3 vaccines-10-00398-f003:**
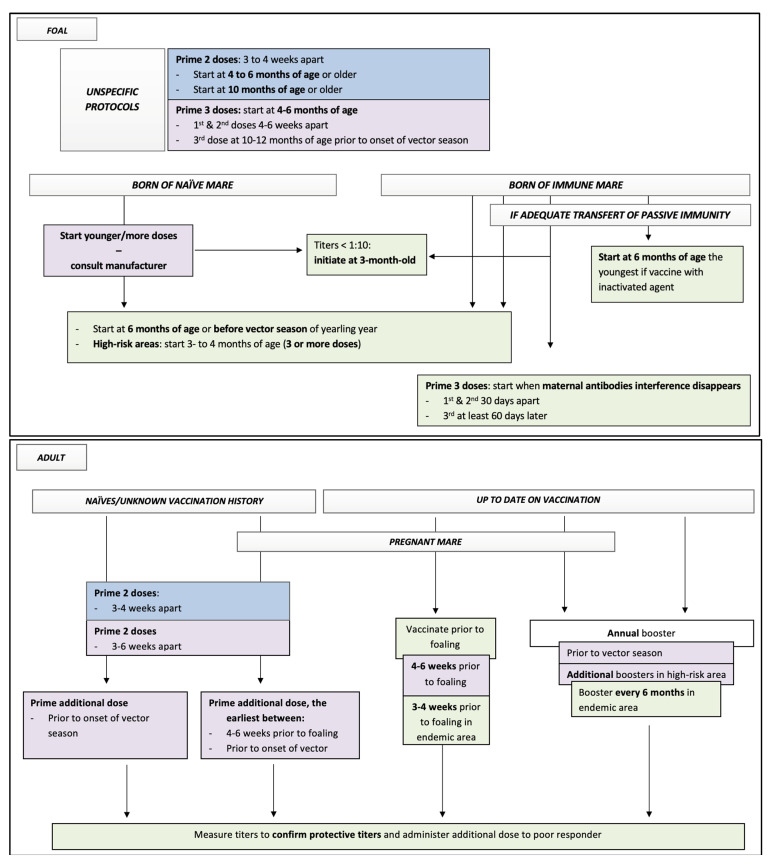
Flow chart presenting all available data to equine practitioners for decision on protocol of vaccination against EEE/WEE in horses. Key: blue squares, data from all labelled protocols; purple squares, data from AAEP recommendations in 2021; green squares, data from equine literature. See tables for details regarding supportive data and specific labeled protocols.

**Figure 4 vaccines-10-00398-f004:**
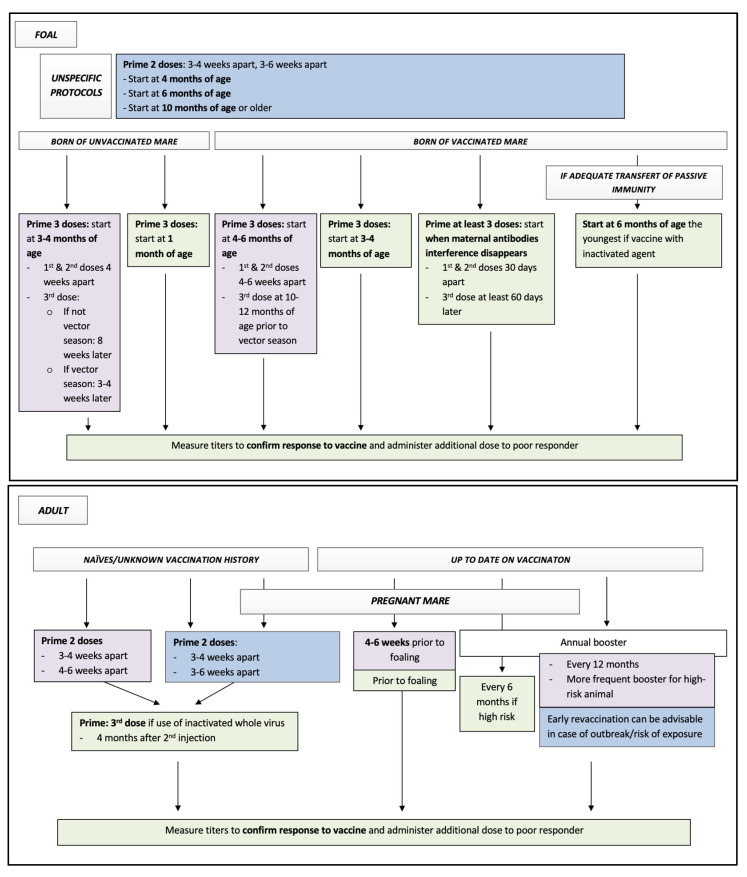
Flow chart presenting all available data to equine practitioners for decision on protocol of vaccination against West Nile in horses. Key: blue squares, data from all labelled protocols; purple squares, data from AAEP recommendations in 2021; green squares, data from equine literature. See tables for details regarding supportive data and specific labeled protocols.

**Table 1 vaccines-10-00398-t001:** Summary of some available protocols and guidelines of tetanus toxoid vaccines in equine in 2021.

Vaccine: AAEP Protocol	Labelled Protocols of Equine Core Vaccine Approved in North America	Suggested Additional Recommendations
Adult - Unvaccinated/unknown vaccination history (including pregnant): ○ Primary vaccination: 2 doses 4–6 weeks apart. ○ If at risk: additionally, tetanus antitoxin at distance of vaccination site when initiating vaccination - Recovered from tetanus or previously vaccinated: Annual booster. ○ If booster >6 months and prior (surgical procedure)/during (wound) exposure. - Pregnant: 4–6 weeks prior foaling. Foal, primary vaccination - 3 dose series: 2 doses 4–6 weeks apart, third dose at 10–12 months of age. - Of vaccinated mares in the peripartum period: start at 4–6 months of age. - Of unvaccinated mare/unknown vaccination history: start at 3–4 months of age	***Monovalent vaccines*** Tetguard™ (Boehringer Ingelheim) - Primary vaccination: 2 doses 30 days apart - Booster ○ Annual ○ Prior to anticipated exposure Tetanus Toxoid/Super-Tet/Prestige^®^ Tetanus (Intervet/Merck) - Primary vaccination 2 doses 3–4 weeks apart - Booster, historically recommended annually Tetanus toxoid (Zoetis) - Primary vaccination: 2 doses: 4–8 weeks apart ○ Administer 1500 units of tetanus antitoxin if injury is present ○ Booster annually ***Multivalent vaccines*** Core EQ Innovator^®^/West Nile-Innovator^®^ + EWT (Zoetis) - Primary vaccination: 1 dose at >10 months of age then 1 dose of West Nile-Innovator + EWT 3–4 weeks after. - Annual booster. Prestige^®^ 3 (previously Encevac T)/Encevac TC-4 with Havlogen/Prestige^®^ 4/Prestige^®^ 5/Prestige^®^ 3 + WNV (previously Encevac T + WNV)/Prestige^®^ 5 + WNV (Intervet/Merck) - Primary vaccination: 2 dose, 3–4 weeks apart - Start at >6 months of age. - Historically, annual revaccination with this product has been recommended. Equi-Jec^®^ 5/Equi-Jec^®^ 6/Equi-Jec^®^ 7/Equi-Jec^®^ WNV + EWT (Boehringer Ingelheim) - Primary vaccination: 2 dose, 3–4 weeks apart - Start at >4 months of age. - Safe to use in pregnant mares. Vetera^®^ EWT/Vetera^®^ VEWT/Vetera^®^ 4XP +WNV/Vetera^®^ 5XP/Vetera^®^ 6XP/Vetera^®^ EWT + WNV/Vetera^®^ VEWT + WNV (Boehringer Ingelheim) - Primary vaccination: 2 dose, 3–4 weeks apart - Start at >4 months of age. - Safe to use in pregnant mares. - Annual booster, prior to vector season.	Pregnant mare: - Vaccinate prior to foaling (expert opinion [16]) ○ Four to 8 weeks prior foaling (AAEP Proceedings [6]) Foal, primary vaccination - Third dose 8–12 weeks after the second dose (AAEP Proceedings [6]) - Of vaccinated mares ○ Do not start inactivated antigens vaccine before 6 months of age (expert opinion [16]) ○ Do not start before 6 months of age (AAEP Proceeding [6]) ○ In the peripartum: do not start before 6-month-old (peer review cohort study [25]) ○ Start when maternal antibodies interference disappears (expert opinion [16], AAEP Proceeding [6]) ○ At least three doses (AAEP Proceeding [6]): ▪ Two injections 30 days apart (expert opinion [16]) ▪ Third dose at least 60 days after (expert opinion [16]) Test serum titer (peer review case reports [31,32]) - To confirm full protection of patient, - To decide on administration of a booster to animals affected by wounds or going under a surgical procedure - To confirm the non-need of vaccine in patient recovered from tetanus

Footnote: AAEP—American Association of Equine Practitioners.

**Table 2 vaccines-10-00398-t002:** Summary of available monovalent vaccines protecting against tetanus in equine.

Infectious Agent and Description	Vaccines Labelled in the USA	Vaccine Description	Reported Efficacy and Safety
*Clostridium tetani* (TT) - Gram positive anaerobic spore-forming bacterium - Synthetize and release a tetanic neurotoxin - Clinical signs: muscular paralysis and spasm, third eye lid prolapse, respiratory impairment, convulsions, and death.	GoatVac T (Durvet)—not in market since 2007 (personal communication with Durvet)	Tetanus toxoids	Efficacy and Safety—Study data were evaluated by USDA-APHIS prior to product licensure and met regulatory standards for acceptance at the time of submission. (https://www.aphis.usda.gov/wcm/connect/3bba1d24-a8c7-4767-a11f-ead24b1b3b64/188-860100.pdf?MOD=AJPERES&CONVERT_TO=url&CACHEID=ROOTWORKSPACE-3bba1d24-a8c7-4767-a11f-ead24b1b3b64-mvhp..F) (accessed on 3 February 2022)
	Tetguard^TM^ (Boehringer Ingelheim)	Purified toxoid Alum precipitated Preservative-Thimerosal	Study data were evaluated by USDA-APHIS prior to product licensure and met regulatory standards for acceptance at the time of submission. No data are published as this study was submitted to USDA-APHIS prior to 1 January 2007, and APHIS only requires publication of data submitted after that date. (https://www.aphis.usda.gov/wcm/connect/7bbeb060-a2c3-45f6-befc-c6dbb31c7d31/165A-860101.pdf?MOD=AJPERES&CONVERT_TO=url&CACHEID=ROOTWORKSPACE-7bbeb060-a2c3-45f6-befc-c6dbb31c7d31-m9P8gXL) (accessed on 3 February 2022)
	Tetanus Toxoid/Prestige^®^ Tetanus (previously Super-Tet) (Intervet/Merck)	Antigen purification system Tetanus toxoid Havlogen^®^ adjuvant Preservative-Thimerosal	Efficacy—demonstrated in laboratory animals (Guinea pigs) according to 9CFR 113.114(c). Satisfactory result is an antitoxin titer of at least 2.0 A.U. per mL for the serum pool. (https://www.aphis.usda.gov/wcm/connect/7bbeb060-a2c3-45f6-befc-c6dbb31c7d31/165A-860101.pdf?MOD=AJPERES&CONVERT_TO=url&CACHEID=ROOTWORKSPACE-7bbeb060-a2c3-45f6-befc-c6dbb31c7d31-m9P8gXL) (accessed on 3 February 2022) Safety—of 552 horses, 298 received 2 doses IM 3 to 4 weeks apart for primary immunization and 254 horses received 1 dose IM. No horses showed muscular swelling, pain, or stiffness. (https://www.aphis.usda.gov/wcm/connect/7bbeb060-a2c3-45f6-befc-c6dbb31c7d31/165A-860101.pdf?MOD=AJPERES&CONVERT_TO=url&CACHEID=ROOTWORKSPACE-7bbeb060-a2c3-45f6-befc-c6dbb31c7d31-m9P8gXL) (accessed on 3 February 2022)
	Tetanus toxoid (Zoetis)	Tetanus toxoid	Efficacy—demonstrated in laboratory animal (guinea pigs) requirements were evaluated by USDA-APHIS prior to product licensure and met regulatory standards for acceptance per 9 CFR 113.114 (date not specified). (https://www.aphis.usda.gov/wcm/connect/0f81ce58-550f-4b18-b406-f8df3e6f2bcb/190-48R521.pdf?MOD=AJPERES&CONVERT_TO=url&CACHEID=ROOTWORKSPACE-0f81ce58-550f-4b18-b406-f8df3e6f2bcb-mCiisji) (accessed on 3 February 2022)

Footnote: TT—Tetanus toxoid, USDA—United States Department of Agriculture, APHIS—Animal and Plant Health Inspection Service, IM—intramuscularly, A.U.—Absorbance Units, CFR—Code of Federal Regulations.

**Table 3 vaccines-10-00398-t003:** Summary of some available protocols and guidelines of rabies vaccines in equine in 2021.

Vaccine: AAEP Protocol	Labelled Protocols of Equine Core Vaccine Approved in North America	Suggested Additional Recommendations
Adult - Primary vaccination: 1 dose - Annual booster - Animal vaccinated and exposed: booster (cf. state regulations) - Pregnant: no labelled vaccine, mares may be vaccinated before breeding or 4 to 6 weeks before foaling. Foal and Weanling, primary vaccination - Starting at 4–6 months of age as per product label - Of vaccinated mares: 2 doses, 4–6 weeks apart - Of unvaccinated mares/unknown vaccination history: 1 dose	***Monovalent vaccines*** Prestige^®^ EquiRab^®^ (Merk) - Primary vaccination: ○ One dose starting at 4 months of age or older ○ Second dose 12 months after - Booster every 14 months Rabisin/Imrab Large Animal (Merial/Boehringer Ingelheim) - Primary vaccination: 1 dose - Start at 3 months of age or older - Annual booster ***Multivalent vaccine*** Core EQ Innovator^®^ (Zoetis) - Primary vaccination: 1 dose at 10 months of age or older ○ But 14-month duration of immunity in 3-months-old horses - Annual booster.	Adult - Primary vaccination in adults, especially if older than 20-year-old, consider ○ Two doses 4 weeks apart (based on peer review cohort studies [28,29,42]) - Broodmares: ○ Vaccinate before breeding (AAEP Proceedings [6,27]) ○ Vaccinate broodmare prior to foaling (expert opinion [16]) Foal: - Start vaccination at 3 to 4 months old (peer review report [43]) - Born of naïve mares: start vaccination at 3 months old (AAEP Proceedings [6]) - Born of immunized mares ○ Immunized before breeding: start at 6 months old, 1 to 2 doses primary vaccination 4 weeks apart (AAEP Proceedings [6]) ○ Do not start the vaccination before 9 months old, when maternal antibodies interference disappears (expert opinion [16]) Assess the humoral response of vaccinated horses. Additional booster in poor responders might be beneficial (based on peer review cohort studies [28,29,42])

Footnote: AAEP—American Association of Equine Practitioners.

**Table 4 vaccines-10-00398-t004:** Summary of some available monovalent vaccines protecting against rabies in equine.

Infectious Agent and Description	Vaccines Labelled in the USA	Vaccine Description	Reported Efficacy and Safety
Rabies Virus - Bullet-shaped RNA virus, Genus Lyssavirus, Family Rhabdoviridae - Neurotropic - Causes fatal neurological disease in mammals. Affected animals show behavioral changes, abnormalities of the cranial and peripheral nerves, with loss of the lower motor neuron and autonomic function.	EquiRab and Prestige EquiRab (Merck)	Inactivated virus + Havlogen^®^ adjuvant Antibiotic-neomycin + polymyxin B Preservative-Thimerosal	Efficacy—Of 37 animals, 26 4-month-old horses vaccinated once IM, and 11 were challenged with Rabies virus 14-months after with no deaths due to Rabies. (https://www.aphis.usda.gov/wcm/connect/577930f8-81c2-426a-8488-a1570b630955/165A-190523.pdf?MOD=AJPERES&CACHEID=ROOTWORKSPACE-577930f8-81c2-426a-8488-a1570b630955-mdG7YyP) (accessed on 3 February 2022) Safety—992 horses with 413 4-month-old animals received 1 dose IM. No animals showed pain or swelling in the injection site
	Rabisin/Imrab Large Animal (Boehringer Ingelheim)	Tetanus toxoid with betapropiolactone + adjuvant aluminum hydroxide Antibiotic-Gentamicin Preservative-Thimerosal	Study data were evaluated by USDA-APHIS prior to product licensure and met regulatory standards for acceptance at the time of submission. Study data, however, are no longer available. https://www.aphis.usda.gov/wcm/connect/577930f8-81c2-426a-8488-a1570b630955/165A-190523.pdf?MOD=AJPERES&CACHEID=ROOTWORKSPACE-577930f8-81c2-426a-8488-a1570b630955-mdG7YyP) (accessed on 3 February 2022)
			Study results applicable to Intramuscular route of administration. Study data were evaluated by USDA-APHIS prior to product licensure and met regulatory standards for acceptance at the time of submission. No data are published as this study was submitted to USDA-APHIS prior to 1 January 2007, and APHIS only requires publication of data submitted after that date.(https://www.aphis.usda.gov/wcm/connect/577930f8-81c2-426a-8488-a1570b630955/165A-190523.pdf?MOD=AJPERES&CACHEID=ROOTWORKSPACE-577930f8-81c2-426a-8488-a1570b630955-mdG7YyP) (accessed on 3 February 2022)

Footnote: RV—Rabies virus, RNA—Ribonucleic acid, USDA—United States Department of Agriculture, APHIS—Animal and Plant Health Inspection Service, IM—intramuscularly.

**Table 5 vaccines-10-00398-t005:** Summary of some available protocols and guidelines of EEE/WEE vaccines in equine in 2021.

Vaccine: AAEP Protocol	Labelled Protocols of Equine Core Vaccine Approved in North America	Suggested Additional Recommendations
Adults - Unvaccinated/unknown vaccination history (including pregnant) ○ Primary vaccination: 2 doses 3–6 weeks apart as per product label ○ Revaccinate prior to onset of vector season ○ Revaccinate the earliest between 4–6 prior to foaling or onset of vector season - Vaccinated (including pregnant) ○ Annual booster prior to vector season (spring). ○ High risk animals/areas: more frequent vaccination ○ Pregnant: 4–6 weeks prior to foaling - Other considerations: high risk animals/areas, consult manufacturer to start earlier or more frequent vaccination Foal regardless vaccination status of the mares - Primary vaccination: 2 dose series 4–6 weeks apart, starting at 4–6 months of age - Third dose at 10–12 months of age prior to onset of vector season - Other considerations—if unvaccinated mare: consult manufacturer to start earlier or more frequent vaccination	Prestige^®^ 3/Prestige^®^ 4/Prestige^®^ 5/Prestige^®^ 3 + WNV/Prestige^®^ 5 + WNV (Intervet/Merck) - Primary vaccination: 2 dose 3–4 weeks apart - Start at >6 months of age. - Historically, annual revaccination with this product has been recommended. Equi-Jec^®^ 5/Equi-Jec^®^ 6/Equi-Jec^®^ 7/Equi-Jec^®^ WNV + EWT (Boehringer Ingelheim) - Primary vaccination: 2 doses 3–4 weeks apart - Start at 4 months of age. - The need for this booster has not been established. - Safe to use in pregnant mares. Vetera^®^ EWT/Vetera^®^ VEWT/Vetera^®^ 4^XP^ +WNV/Vetera^®^ 5^XP^/Vetera^®^ 6^XP^/Vetera^®^ EWT + WNV/Vetera^®^ VEWT + WNV/Vetera Gold^XP^ (Boehringer Ingelheim) - Primary vaccination: 2 doses, 3–4 weeks apart - Start at >4 months of age. - Safe to use in pregnant mares. - Annual booster, prior to vector season. Core EQ Innovator^®^/West Nile-Innovator^®^ + EWT (Zoetis) - Primary vaccination: 1 dose at >10 months of age then 1 dose of West Nile-Innovator + EWT 3–4 weeks after. - Annual booster.	- Adults ○ Up to date on vaccine (peer review reports [54,55]): ▪ Revaccinate every 6 months in endemic area. ▪ Assess the humoral response of vaccinated horses. Additional booster in poor responders might be beneficial. ○ Broodmare prior to foaling (expert opinion [16]) ▪ 3 to 4 weeks prior to foaling (peer review report [55]) - Foals ○ If titers lower than 1:10, initiate primary vaccination at 3 months old (peer review report [55]) ○ Start primary vaccination at 6-month-old or during spring of yearling year (AAEP Proceedings [6]) ○ If risk is high ▪ Start at 3 to 4 months old with three or more dose (AAEP Proceedings [6]) ▪ Administer dose at 3, 4 and 6 months old (peer review report [55]) ○ If adequate passive immunity transfer start the vaccination at 6 months old the youngest if vaccine with inactivated antigens (expert opinion [16]) ○ Primary vaccination to start when maternal antibodies interference disappears, use at least three doses (expert opinion [16]): ▪ Two injections 30 days apart ▪ Third dose at least 60 days after

Footnote: AAEP—American Association of Equine Practitioners.

**Table 6 vaccines-10-00398-t006:** Summary of some available multivalent vaccines protecting against EEE/WEE in equine.

Infectious Agents and Description	Vaccines Labelled in the USA	Vaccine Description	Reported Efficacy and Safety
Eastern Equine Encephalitis Virus (EEE) and Western Equine Encephalitis Virus (WEE) - Genus Alphavirus, Family Togaviridae, single-stranded, positive-sense RNA virus. - Born-mosquito disease with moderate mortality for EEEV and mild mortality for WEEV that causes fever and neurological signs.	Encevac TC-4 (Intervet/Merck; EIV, TT, EEEV, WEEV, WNV)	Killed virus Tetanus toxoid	Efficacy TT—was demonstrated in laboratory animals (guinea pigs) according to 9CFR 113.114(c). Satisfactory result is an antitoxin titer of at least 2.0 A.U. per mL for the serum pool. Efficacy EEEV—was demonstrated in laboratory animals (guinea pigs) according to 9CFR 113.207(b). Satisfactory test result is a Virus Neutralization Titer of ≥1:40 in at least 9 out of 10 vaccinates (2nd stage—at least 17 out of 20 vaccinates). Efficacy WEEV—was demonstrated in laboratory animals (guinea pigs) according to 9CFR 113.207(b). Satisfactory test result is a Virus Neutralization Titer of ≥1:40 in at least 9 of the vaccinates. Efficacy EHV-1—16/32, 11 months of age horses received 2 vaccination IM doses 3 weeks apart were challenged with EHV-1, DA35 strain. 6/16 horses were positive to virus isolation. 1. Efficacy EHV-4—21/36, 6 months of age horses received 2 vaccination IM doses 3 weeks apart were challenged with EHV-4. Only 2 animals had clinical signs. 2.Efficacy EHV-4—16/21, 6 months of age horses received 2 vaccination IM doses 3 weeks apart; were challenged with EHV-4. Horses shed the virus for a minimum of 3 days and a maximum of 14 days, also, 75% presented nasal discharge. Efficacy EIV—18 horses were challenged with A/equine/Kentucky/99 six months post 2nd IM vaccination 3 weeks apart. 14/18 horses showed clinical signs and 12/18 horses 12/18 horses were positive to virus isolation 14 days after challenge. Safety—298 horses received 2 vaccination IM doses 3–4-weeks apart. 96.47% of the horses had no evidence of reactions. (https://www.aphis.usda.gov/wcm/connect/4f1c0212-8505-43ff-add1-e096881ebfac/165a-4855r2.pdf?MOD=AJPERES&CONVERT_TO=url&CACHEID=ROOTWORKSPACE-4f1c0212-8505-43ff-add1-e096881ebfac-mVCsR7z) (accessed on 3 February 2022)
	Prestige^®^3 (Merck; TT, EEEV, WEEV)		
	Prestige^®^3 + WNV (Merck; TT, EEEV, WEEV, WNV)		
	Prestige^®^4 (Merck; EIV, TT, EEEV, WEEV)	Killed virus Tetanus toxoid Antibiotic-Gentamicin Preservatives-Thimerosal	
	Prestige^®^5 (Merck; EIV, EHV-1/4, TT, EEEV, WEEV)		
	Prestige^®^5 + WNV (Merck; TT, EEEV, WEEV, EIV, EHV-1/4, WNV)		
	Vetera EWT (Boehringer Ingelheim; TT, EEEV, WEEV	Inactivated virus Tetanus toxoid Established Carbimmune^®^ adjuvant	Efficacy TT—10 guinea pigs, 6 weeks after the injection, vaccinate serum samples were collected and pooled, then tested for antitoxin content by indirect ELISA. A satisfactory value, which met the requirements per 9 CFR 113.114(c), was achieved. Efficacy EEEV+WEEV+VEEV—10/12 guinea pigs, Serum samples were tested by a plaque reduction, serum neutralization test, 14 to 21 days after the second injection. Vaccinates and controls were evaluated in terms of Eastern and Western equine encephalomyelitis per the criteria in 9 CFR 113.207(b) and the requirements were met. Efficacy EHV-1—20/40, 4–5 months old horses received 2 vaccination IM doses 3 weeks apart and were challenged with EHV-1 15-days the last dose. Only 6 horses had no nasal discharge. Efficacy EHV-4—20/40, 4-months old horses received 2 vaccination IM doses 3 weeks apart and were challenged 4-days after the last dose. 12/20 had mild and 1/20 had moderate clinical signs 1. Efficacy EIV—20/30 hoses, 5–6-months-old received 2 IM vaccination doses 21 days apart and were challenged with Influenza A/eq/Ohio/2003 administered 184 days post-final vaccination. 9/20 horses had clinical signs. 2. Efficacy EIV—18/37 horses, 9–10-months-oldreceived 2 IM vaccination doses 21 days apart and were challenged with Influenza A/eq/Ohio/2003 administered 3 weeks post-final vaccination. 0/18 horses shed virus. Efficacy WNV—20/30 horses, 4–5-months-old received 2 IM vaccination doses 25 days apart and 10/20 horses were challenged 380 days after the 2nd vaccination dose and 10/20 horses were challenged 408 days after the 2nd vaccination dose. 2/20 horses had viremia and 1/20 horses died. Safety all—622 horses vaccinated with 2 IM doses 21 days apart. Only 8/622 horses showed transient injection site swelling. (https://www.aphis.usda.gov/wcm/connect/283444e4-a323-41e3-b741-f2c022f332c9/124_484721.pdf?MOD=AJPERES&CONVERT_TO=url&CACHEID=ROOTWORKSPACE-283444e4-a323-41e3-b741-f2c022f332c9-mCXh75k) (accessed on 3 February 2022)
	Vetera EWT + WNV (Boehringer Ingelheim; TT, EEEV, WEEV, WNV)		
	Vetera VEWT (Boehringer Ingelheim; TT, EEEV, WEEV, VEEV)		
	Vetera VEWT + WNV (Boehringer Ingelheim; TT, EEEV, WEEV, VEEV, WNV)		
	Vetera 4^XP^ + WNV (Boehringer Ingelheim; EIV, TT, EEEV, WEEV, WNV)		
	Vetera 5^XP^ (Boehringer Ingelheim; EIV, EHV-1/4, TT, EEEV, WEEV)		
	Vetera 6^XP^ (Boehringer Ingelheim; EIV, EHV-1/4, TT, EEEV, WEEV, VEEV)		
	Vetera Gold^XP^ (Boehringer Ingelheim; EIV, EHV-1/4, TT, EEEV, WEEV, VEEV)		
	Equi-Jec^®^ WNV+EWT (Boehringer Ingelheim; TT, EEEV, WEEV, WNV)	Inactivated virus Tetanus toxoid Preservative-Formaldehyde Antibiotic-Gentamicin	
	Equi-Jec 7 (Boehringer Ingelheim; TT, EEEV, WEEV, VEEV, EIV, EHV-1/4, WNV)		
	Equi-Jec 6 (Boehringer Ingelheim; TT, EEEV, WEEV, EIV, EHV-1/4, WNV)		
	Core EQ Innovator^®^/West Nile-Innovator^®^ + EWT (Zoetis; RV, TT, WNV, EEE, WEE)	Inactivated virus Tetanus Toxoid Adjuvant-MetaStim^®^	Efficacy (TT, EEE, WEE)—evaluated by USDA-APHIS in guinea pigs prior to product licensure and met regulatory standards for acceptance per 9 CFR 113.207(b)(2). Efficacy WNV—32, 9–11-months-old horses received 2 vaccination doses 3 weeks apart and challenged 12 months after. 1/19 horses were viremic. Efficacy RV—39, 3-month-old horses received 2 IM vaccination doses 3–4 weeks apart. 2/25 animals were positive—The requirements of 9 CFR 113.209 were met. (https://www.aphis.usda.gov/wcm/connect/0f81ce58-550f-4b18-b406-f8df3e6f2bcb/190-48R521.pdf?MOD=AJPERES&CONVERT_TO=url&CACHEID=ROOTWORKSPACE-0f81ce58-550f-4b18-b406-f8df3e6f2bcb-mCiisji) (accessed on 3 February 2022) Safety—682 horses (https://www.aphis.usda.gov/wcm/connect/0f81ce58-550f-4b18-b406-f8df3e6f2bcb/190-48R521.pdf?MOD=AJPERES&CONVERT_TO=url&CACHEID=ROOTWORKSPACE-0f81ce58-550f-4b18-b406-f8df3e6f2bcb-mCiisji) (accessed on 3 February 2022)
	Core EQ Innovator^®^ + V (Zoetis; RV, TT, WNV, EEE, WEE, VEE)		

Footnote: TT—Tetanus toxoid, VEEV—Venezuelan Equine Encephalitis Virus, EEE—Eastern Equine Encephalitis, EEEV—Eastern Equine Encephalitis Virus, WEE—Western Equine Encephalitis, WEEV—Western Equine Encephalitis Virus, WNV—West Nile Virus, EIV—Equine Influenza Virus, EHV-1/4—Equine Herpes Virus 1/4, RNA—Ribonucleic acid, IM—intramuscularly, A.U.—Absorbance Units, CFR—Code of Federal Regulations, ELISA—Enzyme-Linked Immunosorbent Assay.

**Table 7 vaccines-10-00398-t007:** Summary of some available protocols and guidelines of West Nile vaccines in equine in 2021.

Vaccine: AAEP Protocol	Labelled Protocols of Equine core Vaccine Approved in North America	Suggested Additional Recommendations
Adult - Unvaccinated/unknown vaccination history ○ Inactivated whole virus vaccine, recombinant canary pox vaccine: primary vaccination 2 doses 4–6 weeks apart ○ Inactivated flavivirus chimera vaccine: primary vaccination 2 doses 3–4 weeks apart ○ Pregnant: acceptable to vaccinate *, only one vaccine labelled ** - Vaccinated ○ Annual booster (every 12 months) prior to vector season (spring). ○ High risk animals/areas, more frequent vaccination ○ Pregnant: 4–6 weeks prior to foaling Foals - Of vaccinated mares: start at 4–6 months of age ○ Inactivated whole virus vaccine: ▪ Primary vaccination: 2 doses 4–6 weeks apart ▪ Third dose: 10–12 months of age prior to vector season ○ Recombinant canary pox vaccine, inactivated flavivirus chimera vaccine ▪ Primary vaccination: 2 doses 4 weeks apart ▪ Third dose: 10–12 months of age prior to vector season - Of unvaccinated mares ○ Primary vaccination: 2 doses 4 weeks apart at 3–4 months of age ○ Third dose ▪ If not mosquito vector season: 8 weeks apart from the second dose ▪ If during vector season: 3–4 weeks apart from the second dose	***Monovalent vaccines*** West Nile-Innovator^®^ (Zoetis) - Primary vaccination: 2 dose of 3–6 weeks apart - Starting at >10 months of age. - Early revaccination may be advisable when horses are faced with an outbreak or with other conditions that might make exposure likely. - Annual booster. Equi-Jec^®^ WNV (Boehringer Ingelheim) - Primary vaccination: 2 doses 3–4 weeks apart - Start at 4 months of age. - Safe to use in pregnant mares. Vetera^®^ WNV (Boehringer Ingelheim) - Primary vaccination: 2 doses 3–4 weeks apart - Start at 4 months of age. - Safe to use in pregnant mares. - Annual booster, prior to vector season. ***Multivalent vaccines*** Prestige^®^ 3 + WNV (previously Encevac T + WNV)/Prestige^®^ 5 + WNV (Intervet/Merck) - Primary vaccination: 2 doses, 3–4 weeks apart - Start at >6 months of age. - Historically, annual revaccination with this product has been recommended. Equi-Jec^®^ 6/Equi-Jec^®^ 7/Equi-Jec^®^ WNV + EWT (Boehringer Ingelheim) - Primary vaccination: 2 doses 3–4 weeks apart - Start at 4 months of age. - Safe to use in pregnant mares. Vetera^®^ 4^XP^ + WNV/Vetera^®^ 5^XP^/Vetera^®^ 6^XP^/Vetera^®^ EWT + WNV/Vetera^®^ VEWT + WNV (Boehringer Ingelheim) - Primary vaccination: 2 doses 3–4 weeks apart, start at 4 months of age. - Safe to use in pregnant mares. - Annual booster, prior to vector season. Core EQ Innovator^®^/West Nile-Innovator^®^ + EWT (Zoetis) - Primary vaccination: 1 dose at >10 months of age then 1 dose of West Nile-Innovator + EWT 3–4 weeks after. - Annual booster.	Adult - Inactivate whole virus: a third injection 4 months after the primary vaccination might be valuable (peer review reports [52]) - Mare: vaccinate annually once or twice if high risk area (AAEP Proceedings [6]) - Vaccinate broodmare prior to foaling (expert opinion [16]) Foal - Three doses for primary vaccination (AAEP Proceedings [6]) - Born from vaccinated mares: start primary vaccination at 3 to 4 months old (AAEP Proceedings [6]) - Born from non-immunized mares: start primary vaccination as early as 1 month old (AAEP Proceedings [6]) - If adequate passive immunity transfer, start the vaccination at 6 months old the youngest if vaccine with inactivated antigens (expert opinion [16]) - Start when maternal antibodies interference disappears, use at least three doses (expert opinion [16]): ○ Two injections 30 days apart ○ Third dose at least 60 days after Assess the humoral response of vaccinated horses. Additional booster in poor responders might be beneficial (based on peer review report [61])

Footnote: AAEP—American Association of Equine Practitioners; *, unclear if acceptable to use non label product; **, in 2021 more than one of the currently licensed WN vaccines carries a safe for use in pregnant mare label claim.

**Table 8 vaccines-10-00398-t008:** Summary of some monovalent available vaccines protecting against WNV in equine in North America in 2021.

Infectious Agent and Description	Vaccines Labelled in the USA	Vaccine Description	Reported Efficacy and Safety
West Nile Virus (WNV) - Family Flaviviridae, Genus Flavivirus, single-stranded, positive-sense, RNA virus. - The neurologic disease has a moderate case-fatality rate.	West Nile-Innovator^®^ (Zoetis)	Killed virus	Efficacy TT, EEE, and WEE—evaluated by USDA-APHIS in guinea pigs prior to product licensure and met regulatory standards for acceptance per 9 CFR 113.207(b)(2). Efficacy WNV—32, 9–11-months-old horses received 2 vaccination doses 3 weeks apart and challenged 12 months after. 1/19 horses were viremic. Efficacy RV—39, 3-month-old horses received 2 IM vaccination doses 3–4 weeks apart. 2/25 animals were positive—The requirements of 9 CFR 113.209 were met. (https://www.aphis.usda.gov/wcm/connect/0f81ce58-550f-4b18-b406-f8df3e6f2bcb/190-48R521.pdf?MOD=AJPERES&CONVERT_TO=url&CACHEID=ROOTWORKSPACE-0f81ce58-550f-4b18-b406-f8df3e6f2bcb-mCiisji) (accessed on 3 February 2022) Safety—682 horses (https://www.aphis.usda.gov/wcm/connect/0f81ce58-550f-4b18-b406-f8df3e6f2bcb/190-48R521.pdf?MOD=AJPERES&CONVERT_TO=url&CACHEID=ROOTWORKSPACE-0f81ce58-550f-4b18-b406-f8df3e6f2bcb-mCiisji) (accessed on 3 February 2022)
	EquiNile (Intervet/Merck)	Inactivated chimera flavivirus	Efficacy—Of 40 horses, 20 vaccinates were challenged; 10 at 60 days post-2nd vaccination and 10 at 91 days post-2nd vaccination. (https://www.aphis.usda.gov/wcm/connect/5659ee49-dd51-48bb-989f-6bfc5db10e31/165A-1995R2.pdf?MOD=AJPERES&CONVERT_TO=url&CACHEID=ROOTWORKSPACE-5659ee49-dd51-48bb-989f-6bfc5db10e31-m9P7PYj) (accessed on 3 February 2022) Safety—1255 horses received 2-doses 3–4 weeks apart. 1149/1255 horses had no evidence of reactions. (https://www.aphis.usda.gov/wcm/connect/5659ee49-dd51-48bb-989f-6bfc5db10e31/165A-1995R2.pdf?MOD=AJPERES&CONVERT_TO=url&CACHEID=ROOTWORKSPACE-5659ee49-dd51-48bb-989f-6bfc5db10e31-m9P7PYj) (accessed on 3 February 2022)
	PreveNile (Merck)	Live attenuated chimeric vaccine—No longer available	No longer available
	Vetera WNV (Boehringer Ingelheim)	Inactivated virus + Established Carbimmune^®^ adjuvant	1. Efficacy—19/28 horses, 4–5-months-old were vaccinated with 2 IM doses 21 days apart. 10/19 horses were challenged intrathecally 14 days after 2nd vaccination dose and 9/19 horses were challenged intrathecally 28 days after 2nd vaccination dose. 1/19 horses was viremic. 2. Efficacy—20/30 horses were vaccinated with 2 doses 25 days apart. 10/20 were challenged 380 days after 2nd vaccination and 10/20 were challenged 408 days after 2nd vaccination dose with WNV. 1/20 vaccinates died and 2/20 vaccinates had viremia. 3. Efficacy—20/30 horses, 4–5-months-old were vaccinated with 2 IM doses 25 days apart. 10/20 horses were challenged 380 days after 2nd vaccination dose and 10/20 horses were challenged 408 days after 2nd vaccination dose. 2/20 horses had viremia and 1/20 died. Safety—325 pregnant mares were vaccinated with 2 IM doses 16–28 days apart, during 1st, 2nd, and 3rd trimester. No evidence of disease, abortion or death due to vaccination. Safety all—556 horses received 2 vaccination IM doses 3–4 weeks apart. 99.3% of horses after 1st dose showed no reactions and 98% of horses after the 2nd dose showed no reactions. (https://www.aphis.usda.gov/wcm/connect/aafa450c-43ea-476c-bfdc-ab3b2638486b/124_199520.pdf?MOD=AJPERES&CONVERT_TO=url&CACHEID=ROOTWORKSPACE-aafa450c-43ea-476c-bfdc-ab3b2638486b-nKD-5yM) (accessed on 3 February 2022)
	Equi-Jec^®^ WNV (Boehringer Ingelheim)	Preservative-Formaldehyde Antibiotic-Gentamicin	

Footnote: TT—Tetanus toxoid, EEE—Eastern Equine Encephalitis, WEE—Western Equine Encephalitis, WNV—West Nile Virus, RV—Rabies virus, RNA—Ribonucleic acid, IM—intramuscularly, CFR—Code of Federal Regulations

## Data Availability

Not applicable.

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
