# Peer review of "Equids’ Core Vaccines Guidelines in North America: Considerations and Prospective"

_vaccines, 2022, doi:10.3390/vaccines10030398_

Round 1

Reviewer 1 Report

This is a very well written paper describing focused present potential limitations of the AAEP guidelines equine core vaccinations and resources that can complete these guidelines. Authors also propose future research to improve the guidelines of equine core vaccination.  In my opinion present paper will help equine practitioners in better understanding the benefits and limitation of core vaccines and the scientific background of vaccination protocols.  

1. The table of vaccines could be advisable.
2. Other addition may be a graphical workflow of decision making process when and what kind of vaccine to use.

Author Response

We would like to thank the reviewer for the comments and suggestion. By addressing these, we hope the editor will consider publishing our findings, as all reviewers seem to agree that the manuscript is contributing to the field.

For ease of reading, we italicized and bolded each reviewer’s comments, followed by our response.

This is a very well written paper describing focused present potential limitations of the AAEP guidelines equine core vaccinations and resources that can complete these guidelines. Authors also propose future research to improve the guidelines of equine core vaccination.  In my opinion present paper will help equine practitioners in better understanding the benefits and limitation of core vaccines and the scientific background of vaccination protocols.  

  1. The table of vaccines could be advisable.

Table 1 is presenting the equine core vaccines of North America along with the protocols approved for each label.

We built other tables as suggested by a reviewer.

If the tables and flow charts added are not sufficient, could the reviewer specify what type of table of vaccines is requested.

  1. Other addition may be a graphical workflow of decision making process when and what kind of vaccine to use.

This has been implemented in the revised manuscript.

Reviewer 2 Report

There are some important and interesting facts included in this paper. The paper is difficult to read as there is only one table and no figures. It would help if the authors include more tables and even figures. One table I suggest is one that has the first column "Infectious Agent" that is followed by "Description", "Vaccines Available", "Vaccine Description and Known Efficacy".  The "Description" should include description of the infectious agent e.g. RNA virus, Alphavirus and virulence. A table like that would go a long way to making the paper for readable.

Author Response

We would like to thank the reviewer for the comments and suggestion. By addressing these, we hope the editor will consider publishing our findings, as all reviewers seem to agree that the manuscript is contributing to the field.

For ease of reading, we italicized and bolded each reviewer’s comments, followed by our response.

English language and style are fine/minor spell check required

We implemented correction of English language in the revised manuscript and had a proofreader who checked before submission. The proof reader use the book called “Strunk and white” as rationales.

However, the authors had the manuscript proof-read by 3 American native speakers before initial submission. Two of the proofreaders have veterinary medical background, with one with equine and research background, and the other a small animal background; one proofreader has no medical background but a science background. Two of these proofreaders have also published paper in peer review journals. If the reviewer wants additional corrections, can we get more details on spell check required, or guidance on the type of proofreaders that should be contacted to refine the manuscript.

There are some important and interesting facts included in this paper. The paper is difficult to read as there is only one table and no figures. It would help if the authors include more tables and even figures. One table I suggest is one that has the first column "Infectious Agent" that is followed by "Description", "Vaccines Available", "Vaccine Description and Known Efficacy".  The "Description" should include description of the infectious agent e.g. RNA virus, Alphavirus and virulence. A table like that would go a long way to making the paper for readable.

Tables were built as suggested has been added in the revised manuscript.

Reviewer 3 Report

The authors present an interesting review that explores the currently recommended vaccinations for equines in North America. They have compared and contrasted the comparatively slow development of these recommendations for equines compared to that of companion animals. They have also identified where additional research could be done to support the use of vaccines. Overall the review is interesting and should be of interest to those working to improve the health of equines. There is a risk of limited readership due to the focus on the situation in North America, but at the same time, the authors have drawn on data from across the globe. Thus, it should have some relevance to those in other jurisdictions.

In many places throughout the manuscript, the authors review the relevant literature and then suggest “additional research is required”, I would suggest they consider briefly expanding these statements to suggest the type of research required. Similarly, in respect to some of the studies reviewed, the authors do not take regarding the robustness of the study in question. I would suggest they carefully review the included studies in this context.

I have made some additional suggestions below for the authors to consider.

Line 12 suggest revision - “In the North American equine medicine, vaccine”

Line 21 suggest revisions - “for small companion animals and horses”

Consider adopting this change throughout the manuscript where the term “small animals” is used.

Line 41 suggest revision - “immunization has proven”

As this is a review of the current literature the tense should be mostly “past”.

Line 41 suggest revision - “humoral responses resulting the production of pathogen specific antibodies.”

I would also strongly argue that live-attenuated vaccines stimulate cell-mediated responses. The statement would apply to inactivated and subunit vaccines, not live-attenuated vaccines. Similarly, not all antibody responses act to neutralise the pathogen. This might be the case for the majority of vaccines that target viruses, but for other pathogens other immune pathways are more important. For example, opsonisation is important for some bacteria.

Line 142-143 – The authors comment on an apparent increased risk of disease in pregnant mares as resulting from vaccination. For such a potentially controversial statement, I would suggest that authors consider expanding this section, not just to report the statement of another study, but to explore what the underlying evidence is for this statement. Addressing such things as the robustness of the data which underpins such a statement is the critical component of a review such as this.

Line 149 Here the authors state: “Finally, many vaccines contain thimerosal, an agent that has raised controversial safety concerns for the fetus both in humans and animals [3].”

Given the importance of such a statement and the surrounding body of literature, I would have expected reference to be made to more recently published studies. Noting that at least some of the literature surrounding thimerosal has been retracted and/or remains unverified following robust peer review. That is it is a controversial area but still needs to delt with appropriately.

Line 153 What is “EWT”?

Apologies if I have missed it but it does not appear to be in the abbreviation list and the relevant links from Table 1 do not work correctly.

Line 173 please review the statement below as I am not sure what it means,

“In addition, the AAEP guidelines lack recommendations to vaccinate foals born from mares updated on vaccines but not vaccinated during peripartum period.”

Line 194 Do the authors mean “prime” rather than “primo” – here and elsewhere?

Line 258 – what do the authors mean by third generation vaccines?

Line 274 – how many horses were included in this study? While 7% of horses were below the level of protection, 93% of horses were above it.

Line 288 New Zealand should not be hyphenated.

Line 342 If the study was not accessible, how was it reviewed to enable inclusion in the review?

Line 373-376 – the authors contrast two studies Ref 41 and Ref 43 – however one relates to cell-mediated responses and the other humoral responses. It is unclear what the significance of this comparison is. Please review.

Line 438 – The important information missing here is the basis of how the eight strains of WNV have been differentiated. If it is based on genotyping (i.e. molecular rather than antigenic) would the authors expect there to be cross-protection between the vaccine strain and the other strains? Has this been investigated? Would the authors consider this to be an important future research direction?

Line 627 suggest revision “safety in equines”

Table 1 – many of the links in the footnote of the table do not work correctly or go to a page that requires log-in credentials for veterinary practitioners. For the purposes of this review, the authors should ensure that all links work and go to generally accessible pages.

Author Response

We would like to thank the reviewer for the comments and suggestion. By addressing these, we hope the editor will consider publishing our findings, as all reviewers seem to agree that the manuscript is contributing to the field.

For ease of reading, we italicized and bolded each reviewer’s comments, followed by our response.

Moderate English changes required

We implemented correction of English language in the revised manuscript and had a proofreader who checked before submission. The proofreader used the book called “Strunk and white” as rationales.

However, the authors had the manuscript proof-read by 3 American native speakers before initial submission. Two of the proofreaders have veterinary medical background, with one with equine and research background, and the other a small animal background; one proofreader has no medical background but a science background. Two of these proofreaders have also published paper in peer review journals. If the reviewer wants additional corrections, can we get more details on spell check required, or guidance on the type of proofreaders that should be contacted to refine the manuscript.

The authors present an interesting review that explores the currently recommended vaccinations for equines in North America. They have compared and contrasted the comparatively slow development of these recommendations for equines compared to that of companion animals. They have also identified where additional research could be done to support the use of vaccines. Overall the review is interesting and should be of interest to those working to improve the health of equines. There is a risk of limited readership due to the focus on the situation in North America, but at the same time, the authors have drawn on data from across the globe. Thus, it should have some relevance to those in other jurisdictions.

In many places throughout the manuscript, the authors review the relevant literature and then suggest “additional research is required”, I would suggest they consider briefly expanding these statements to suggest the type of research required.

In the conclusion, we summarized the topics of research that should be performed. We specified the type of research we suggest in the revised manuscript.   

Similarly, in respect to some of the studies reviewed, the authors do not take regarding the robustness of the study in question. I would suggest they carefully review the included studies in this context.

I have made some additional suggestions below for the authors to consider.

Line 12 suggest revision - “In the North American equine medicine, vaccine”

The sentence has been edited in the revised manuscript.

Line 21 suggest revisions - “for small companion animals and horses”

This has been edited in the revised manuscript.

Consider adopting this change throughout the manuscript where the term “small animals” is used.

These have been edited in the revised manuscript.

Line 41 suggest revision - “immunization has proven”

This has been corrected in the revised manuscript.

As this is a review of the current literature the tense should be mostly “past”.

We changed most tense in the revised manuscript. The manuscript was proof read before submission.

Line 41 suggest revision - “humoral responses resulting the production of pathogen specific antibodies.”

This has been revised.

I would also strongly argue that live-attenuated vaccines stimulate cell-mediated responses. The statement would apply to inactivated and subunit vaccines, not live-attenuated vaccines. Similarly, not all antibody responses act to neutralise the pathogen. This might be the case for the majority of vaccines that target viruses, but for other pathogens other immune pathways are more important. For example, opsonisation is important for some bacteria.

We agree with the reviewer. However, we do not find where we state that live-attenuated vaccines stimulate cell-mediated responses. In the introduction, we mention that “Inactivated and live-attenuated vaccines stimulate humoral responses by synthesis of neutralizing antibodies.”

Line 142-143 – The authors comment on an apparent increased risk of disease in pregnant mares as resulting from vaccination. For such a potentially controversial statement, I would suggest that authors consider expanding this section, not just to report the statement of another study, but to explore what the underlying evidence is for this statement. Addressing such things as the robustness of the data which underpins such a statement is the critical component of a review such as this.

We have commented the section in the revised manuscript and state the level of evidence for this statement.

Line 149 Here the authors state: “Finally, many vaccines contain thimerosal, an agent that has raised controversial safety concerns for the fetus both in humans and animals [3].”

Given the importance of such a statement and the surrounding body of literature, I would have expected reference to be made to more recently published studies. Noting that at least some of the literature surrounding thimerosal has been retracted and/or remains unverified following robust peer review. That is it is a controversial area but still needs to delt with appropriately.

A more recent publication is referenced in the revised manuscript.

Line 153 What is “EWT”?

Apologies if I have missed it but it does not appear to be in the abbreviation list and the relevant links from Table 1 do not work correctly.

The abbreviation is now listed in the revised manuscript. The links have been checked and work on our documents. We would need guidance from the editor for how to make them work for the reviewers.

Line 173 please review the statement below as I am not sure what it means,

“In addition, the AAEP guidelines lack recommendations to vaccinate foals born from mares updated on vaccines but not vaccinated during peripartum period.”

We rephrased the sentence in the revised manuscript. We want to highlight that maybe we should differentiate protocols for foals born from:

  • Mares up to date on their vaccination but NOT receiving additional booster during pregnancy
  • And mares up to date on their vaccination but receiving additional booster during pregnancy in the objective to enhance protection of the foal from passive immunity

Line 194 Do the authors mean “prime” rather than “primo” – here and elsewhere?

We edited all the revised manuscript changing “primo” to “primary” as used by AAEP.

Line 258 – what do the authors mean by third generation vaccines?

We define third-generation vaccine in the introduction: “Third-generation vaccines, DNA, RNA and recombinant viral-vector, also stimulate T-cell mediated immune responses, suggested to be at least as important than the humoral response to protect against infection [1]”

Line 274 – how many horses were included in this study? While 7% of horses were below the level of protection, 93% of horses were above it.

This sentence has been removed in the revised manuscript as the threshold used was 0.1IU/mL instead of 0.01IU/mL.

Line 288 New Zealand should not be hyphenated.

This has been edited in the revised manuscript.

Line 342 If the study was not accessible, how was it reviewed to enable inclusion in the review?

The beginning of the paragraph was changed, and this study is not mentioned in the revised manuscript.

Line 373-376 – the authors contrast two studies Ref 41 and Ref 43 – however one relates to cell-mediated responses and the other humoral responses. It is unclear what the significance of this comparison is. Please review.

The paragraph has been edited to improve clarity in the revised manuscript.

Line 438 – The important information missing here is the basis of how the eight strains of WNV have been differentiated. If it is based on genotyping (i.e. molecular rather than antigenic) would the authors expect there to be cross-protection between the vaccine strain and the other strains? Has this been investigated? Would the authors consider this to be an important future research direction?

A comment to drive attention to molecular vs antigenic concepts is added in the revised manuscript.

However, to avoid confusion, we changed the term “strains” to “lineages” as references  stated. As there are 8 lineages, with multiple strains per lineages, we believe this would be a more accurate statement (Vazquez A, Sánchez-Seco MP, Ruiz S, Molero F, Hernández L, Moreno J, Magallanes A, Tejedor CG, Tenorio A. 2010. Putative new lineage of West Nile virus, Spain. Emerg Infect Dis 16:549–552. http://dx.doi.org/10.3201/eid1603.091033). In horses, lineages 1 and 2 are the only ones pathogenic (cf. manuscript for references). But we cannot find more insight regarding the different strains of WNV and their antigenic differences in horses.

We mention that there is evidence of cross-protection between lineage 1 and 2 in horses: “Labeled vaccines have been developed with lineage 1, but cross protection has been demonstrated against lineage 2 strains.”). We also mention the cross protection to WNV from vaccination to other Flavivirus. Overall, we suggested further investigation regarding cross protection to Flavivirus in horses.

Line 627 suggest revision “safety in equines”

This has been edited in the revised manuscript.

Table 1 – many of the links in the footnote of the table do not work correctly or go to a page that requires log-in credentials for veterinary practitioners. For the purposes of this review, the authors should ensure that all links work and go to generally accessible pages.

The links have been edited in the revised manuscript.

Reviewer 4 Report

The manuscript would be a review of lecterature on the Vaccination in horse. The scope of the manuscript is given to the equine practitioners  a guideline for the vaccination of the horses, considering that these information are available only on American Association of Equine Practitioners (AAEP) website. This consideration is a limit of this study that give information only for American equine praticioners. The description of the various vaccination are not contestualized for an international reader.

On the basis of its limits the manuscript could be consider for pubblication after major revision.

In the title in America should be added

Abstract 

Lines 17-19 should be deleted

Line 20. The authors talk about small animals, but in the title they refer only to equine

Introduction

Lines 68-73 should be deleted

Lines 55-66 the authors refer only to AAEP, in other country there are other guidelines

Paragraph 2 . Establishment of guidelines in dogs and cats should be deleted

The AAEP guidelines are too enphatised 

Paragraph 3 should be shortened or deleted, the information should be put in the other paragraph referendum to the specific deseases.

Rabies, in Europe it is not praticate in horses, it shoukd be better explained  the necessity of this vaccine in horses.

Equine Arboviroses, how can horses infected with this virus, i? In which situation is better to use this vaccine?

Wnd, refer to Arfuso et al.,J equjne vet sci, 2021, 11, 477

Conclusion

Line 616 the research are.... should be rewrittem

Lines 648-652 should be deleted

The conclusion section is too long and should better describe the statement 

Author Response

We would like to thank the reviewer for the comments and suggestion. By addressing these, we hope the editor will consider publishing our findings, as all reviewers seem to agree that the manuscript is contributing to the field.

For ease of reading, we italicized and bolded each reviewer’s comments, followed by our response.

The manuscript would be a review of lecterature on the Vaccination in horse. The scope of the manuscript is given to the equine practitioners  a guideline for the vaccination of the horses, considering that these information are available only on American Association of Equine Practitioners (AAEP) website. This consideration is a limit of this study that give information only for American equine praticioners. The description of the various vaccination are not contestualized for an international reader.

On the basis of its limits the manuscript could be consider for pubblication after major revision.

In the title in America should be added

“North America” has been added in the revised manuscript to reflect other comments from the reviewer (see below).

Abstract 

Lines 17-19 should be deleted

Line 20. The authors talk about small animals, but in the title they refer only to equine

In the title, we stated “considerations” regarding the core equine vaccine. By definition (Oxford dictionary), consideration refers to “a fact or a motive taken into account in deciding or judging something”. The review aims to present and discuss multiple facts about vaccination guidelines of core vaccines in equines:

  • How guidelines have been established in another veterinary field (in small animal): to justify that the AAEP guidelines have limitations in presenting evidence-based guidelines
  • The literatures on equine core vaccines: to highlight some considerations regarding the AAP guidelines.

This is why only the word “considerations” is present in the title. However, we would consider any suggestions to introduce the concept of small animals in the title, but we would consider important to also keep the fact that most of the review is about considerations from the equine literature and the AAEP recommendations.

As “summarize how guidelines are established for small animals to serve as a base of comparison with equine vaccination guidelines” is an aim of the review, and no other reviewers suggested to remove the small animal part, we would prefer not to remove the sentences referring to the small animal field.

We edited in the revised manuscript the parts about small animals in order to improve clarity of the review.

Introduction

Lines 68-73 should be deleted

In line of our response above, we decided to keep these sentences.

Lines 55-66 the authors refer only to AAEP, in other country there are other guidelines

In North America, we are not aware of other exhaustive open access guidelines as offered by the AAEP in Canada. Similarly, we did not find an equivalent of AAEP for all European countries. If we take the example of France on the equine association (AVEF: https://avef.fr/documentations-et-ressources/), no vaccine guidelines are found available for the equine practitioners, and on the French Veterinary Medical Association (https://www.veterinaire.fr/je-suis-veterinaire/mon-exercice-professionnel/les-fiches-professionnelles/la-prevention-contre-la-rage), no vaccine guideline are found either. In the UK, we suspect that NOAH would be the reference. The vaccination topic in equids is short (https://www.noah.co.uk/briefingdocument/equine-vaccination/). Finally, in Germany, the guideline seems to be the “Leitlinie zur Impfung von Pferden”, that can be found online at https://www.bundestieraerztekammer.de/tieraerzte/leitlinien/downloads/Impfleitlinie-Pferde_2019-02-01.pdf  as referenced in Recknagel et al., Pferdeheilkunde 31 (2015). So, if we agree that there are guidelines in other countries, we believe information is not easily accessible to practitioner, or at least not as easy/accessible as the AAEP guidelines. Therefore, as we decided to focus the review on the most accessible information for all English speaker equine practitioner, we refer only to the AAEP.

Furthermore, as suggested by the reviewer, we edited the title to highlight that the review focuses North America guidelines, and therefore on the AAEP guidelines.

Paragraph 2 . Establishment of guidelines in dogs and cats should be deleted

In line of our above response, we would keep this paragraph.

The AAEP guidelines are too enphatised 

As suggested, the title is edited, and the focus is driven towards North American equine practitioners.

Paragraph 3 should be shortened or deleted, the information should be put in the other paragraph referendum to the specific deseases.

Paragraph 3 presents data and approaches concepts that either include multiple of the different diseases developed thereafter, or vaccines other than the core vaccine (e.g. EVA, Keyhole Limpet), to present considerations in:

  • Vaccination of broodmare
  • Vaccination of foals
  • Vaccination of old/elderly horses

As no other reviewers suggested to remove the paragraph, we would like to leave it. However, we edited it to improved its clarity in the revised manuscript.

Rabies, in Europe it is not praticate in horses, it shoukd be better explained  the necessity of this vaccine in horses.

As suggested by reviewer, we focus the review on North America core vaccine. We believe that adding detail son solely rabies regarding the European situation would bring some confusion.

Furthermore, in the introduction, we define how a vaccine is considered to be core: “Vaccines protecting from endemic diseases, regulated diseases for public health, highly contagious diseases, and severe diseases are defined as core vaccines by the American Veterinary Medical Association”. Non-American readers can consult the local authorities’ information to get the list of diseases present in the area of practice that reach the criteria for being core vaccination.

Equine Arboviroses, how can horses infected with this virus, i? In which situation is better to use this vaccine?

 It is detailed in the revised manuscript.

Wnd, refer to Arfuso et al.,J equjne vet sci, 2021, 11, 477

We do not find an article from Arfuso et al. in the Journal of Equine Veterinary Sciences. Can the reviewer confirm that the article suggested is: “Modulation of Serum Protein Electrophoretic Pattern and Leukocyte Population in Horses Vaccinated against West Nile Virus. Animals 2021, 11, 477. https://doi.org/10.3390/ani11020477”.

In this article, there is an interesting point mentioned in the introduction: “in response to vaccination, acute phase proteins, can be used as reliable biomarkers for predicting the immune memory and vaccine efficacy [16]”. However, the ref. 16 (Vaccination elicits a prominent acute phase response in horses, The Vet. J. 191 (2012) 199-202), is a single study on a specific equine vaccine (tetanus-influenza vaccine), which solely report a prominent increase of acute phase proteins following vaccination. We discuss these findings in the revised manuscript.

Conclusion

Line 616 the research are.... should be rewrittem

This sentence is rewritten in the revised manuscript.

Lines 648-652 should be deleted

In line of our above response, we would keep the paragraphs about small animals.

The conclusion section is too long and should better describe the statement 

We created a paragraph for other consideration and shortened the conclusion, and statements in the conclusion section have been edited in the revised manuscript.

Round 2

Reviewer 2 Report

Improvements seen

Author Response

We would like to thank the reviewer for the comment. 

Reviewer 3 Report

The authors have done an excellent job in the revision of thier manuscript. They have adequately addressed all of the comments and suggestions I made during my initial review. I have no further comments.

Author Response

(The authors gave the same response as above.)

Reviewer 4 Report

I am sorry, but the authors have not improved their manuscript following the reviewer's suggestion. 

Author Response

We would like to thank the reviewer for the comments and suggestions. 

However, we believe we addressed a lot of suggestions and comments.

In the revised manuscript, in the paragraph “other considerations” and the conclusion, we mention the existence of these European/International codes, and elaborate in guidelines in Europe. We hope that it would be considered to be sufficient by the reviewer.

Sincerely